# Exploring the decision-making process in model development: focus on the Arctic snowpack

Cecile B. Menard[1], Sirpa Rasmus[2,3], Ioanna Merkouriadi[4], Gianpaolo Balsamo[5], Annett Bartsch[6], Chris Derksen[7], Florent Domine[8], Marie Dumont[9], Dorothee Ehrich[10], Richard Essery[1], Bruce C. Forbes[2], Gerhard Krinner[11], David Lawrence[12], Glen Liston[13], Heidrun Matthes[14], Nick Rutter[15], Melody Sandells[15], Martin Schneebeli[16], Sari Stark[2]

[1] School of Geosciences, University of Edinburgh, Edinburgh, United Kingdom

[2] Arctic Centre, University of Lapland, Rovaniemi, Finland

[3] Faculty of Biological and Environmental Sciences, University of Helsinki, Helsinki, Finland

[4] Finnish Meteorological Institute, Helsinki, Finland

[5] European Centre for Medium Range Weather Forecasts, Reading, United Kingdom

[6] b.geos, Korneuburg, Austria

[7] Climate Res Div, Environm & Climate Change Canada, Toronto, Canada

[8] Takuvik International Laboratory, Université Laval and CNRS, Quebec City, Canada

[9] France Univ. Grenoble Alpes, Université de Toulouse, Météo-France, CNRS, CNRM, Centre d'Etudes de la Neige, Grenoble, France

[10] Dept Arctic & Marine Biol, UiT Arctic Univ Norway, Tromso, Norway

[11] Inst Geosci Environm, Univ Grenoble Alpes, CNRS, Grenoble, France

[12] National Center for Atmospheric Research, Boulder, USA

[13] Cooperat Inst Res Atmosphere, Colorado State Univ, Ft Collins, USA

[14] Alfred Wegener Institute, Helmholtz Centre for Polar and Marine Research, Potsdam, Germany

[15] Dept Geog & Environm Sci, Northumbria University, Newcastle upon Tyne, England

[16] WSL Institute for Snow and Avalanche Research (SLF), Davos Switzerland

*Correspondence to*: Cecile B. Menard (cecile.menard@ed.ac.uk)

**Abstract.** The Arctic poses many challenges to Earth System and snow physics models, which are commonly unable to simulate crucial Arctic snowpack processes, such as vapour gradients and rain-on-snow-induced ice layers. These limitations raise concerns about the current understanding of Arctic warming and its impact on biodiversity, livelihoods, permafrost and the global carbon budget. Recognizing that models are shaped by human choices, eighteen Arctic researchers were interviewed to delve into the decision-making process behind model construction. Although data availability, issues of scale, internal model consistency, and historical and numerical model legacies were cited as obstacles to developing an Arctic snowpack model, no opinion was unanimous. Divergences were not merely scientific disagreements about the Arctic snowpack, but reflected the broader research context. Inadequate and insufficient resources, partly driven by short-term priorities dominating research landscapes, impeded progress. Nevertheless, modellers were found to be both adaptable to shifting strategic research priorities - an adaptability demonstrated by the fact that interdisciplinary collaborations were the key motivation for model development - and anchored in the past. This anchoring and non-epistemic values led to diverging opinions about whether existing models were "good enough" and whether investing time and effort to build a new model was a useful strategy when addressing pressing research challenges. Moving forward, we recommend that both stakeholders and modellers be involved in future snow model intercomparison projects in order to drive developments that address snow model limitations currently impeding progress in various disciplines. We also argue for more transparency about the contextual factors that shape research decisions. Otherwise, the reality of our scientific process will remain hidden, limiting the changes necessary to our research practice.

## 1 Introduction

If the number of mentions in Intergovernmental Panel on Climate Change Assessment Reports (IPCC AR) can be used as a proxy to quantify the importance of a component in the climate system, then our understanding of the key role played by the cryosphere can be dated to the mid-2000s. Cryosphere processes and feedback covered just 5 pages in the IPCC Working Group 1 (WG1) AR3 (IPCC, 2001), but a 48-page dedicated chapter in the IPCC WG1 AR4 (IPCC, 2007). By the Sixth Assessment Cycle, an IPCC Special Report focused on the role of changing oceans and cryosphere under a changing climate (IPCC, 2019). The average number of mentions per page of the words "Arctic" and "snow" in thirty-one years of IPCC WG1 AR trebled (Fig. 1). Meanwhile, the Arctic as a whole has warmed at twice, with some regions almost four times, the global rate (e.g. Serreze et al., 2000; ACIA, 2005; Walsh, 2014; Rantanen et al., 2022).

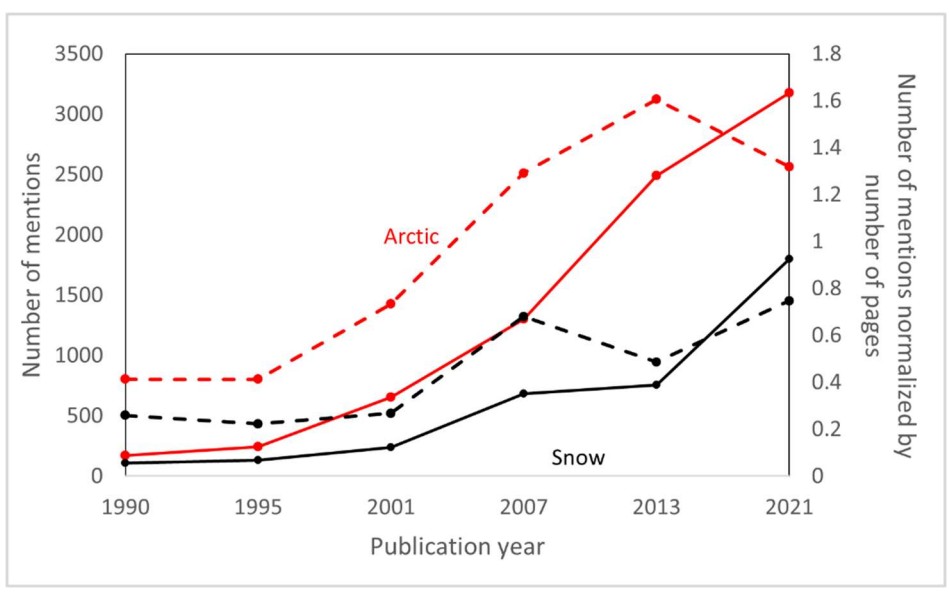

60

*Figure 1: Number of mentions of the words "arctic" (red) and "snow" (black) in each IPCC WG1 AR (IPCC,*

*1990; IPCC, 1995; IPCC, 2001; IPCC, 2007; IPCC, 2013; IPCC, 2021) (solid line) and number of mentions*

*normalized by the number of pages in each report (dashed line).*

The attribution and quantification of climate change by the IPCC WG1 is partly based upon simulations
provided by Earth System models (ESMs), which are lines of code, written over time by multiple scientists, that
describe processes relevant to life on Earth. Other types of models are dedicated to investigating specific
components of the Earth system e.g snow physics models. In both types of models, the "real world" must be
translated into a numerical language, requiring modellers to make decisions at every stage of the model
development. Given limited computing capabilities, modellers must decide which processes matter enough to be
represented, which parametrization of the chosen processes best suits the purpose of their model, which
language to use, how to select or tune parameter values, how to solve the equations, which input data are used,
which decisions to leave to users, which metrics to evaluate their model against; the list of "*the choreography of*
*coded procedures*" (Gramelsberger, 2011) goes on.

The representation of snow in ESMs and snow physics models (hereafter, when combined, referred to as "snow
models") can take on various levels of complexity (here meaning incorporating increasing number of processes)
(see e.g. Slater et al., 2001; Largeron et al., 2020). The simplest representation is a soil-snow composite layer in
which the top soil layer "becomes" snow by adopting some of its attributes when present e.g. albedo, thermal
conductivity. The next complexity level represents a single snow layer where bulk snowpack properties e.g.,
snow water equivalent (SWE), depth and density, are simulated. Finally, multi-layer snow models usually allow
a pre-determined maximum number of snow layers, although some models add snow layers corresponding to
each snowfall, with their specific thickness, density and other attributes.

Most multi-layer snow models use a densification model first developed by Anderson (1976), itself based on
measurements made by Kojima (1967) in Sapporo and Moshiri, Hokkaido, Japan (hereafter the Anderson-
Kojima scheme). The model parameters account for compaction due to the weight of the overlying snow, as

well as destructive, constructive and melt metamorphism; as such, each layer increases in density with depth.
This snow profile broadly resembles the properties associated with montane forest and maritime snow (Sturm
and Liston, 2021), but is not appropriate to simulate wind-packed snow and depth-hoar, i.e. what Arctic tundra
snowpacks are often almost entirely composed of (Fig. 2). Some snow physics models attempt to simulate
Arctic-specific snowpack processes: the vapour diffusion that leads to depth hoar formation, the internal
snowpack ice layers that commonly occur after rain-on-snow events, the thick ice crust that forms at the surface
of the snowpack following freezing rain (e.g. SNOWPACK in Wever et al., 2016 and Jafari et al., 2020;
SnowModel in Liston et al., 2020; Crocus in Quéno et al., 2018, Touzeau et al., 2018 and Royer et al., 2021).
However ,no ESM, i.e. none of the state of the art models that are used by researchers and policymakers
globally to understand the complex interactions in the Earth's climate system, so far, simulates Arctic-specific
snowpack processes. This is despite many in the climate change scientific community considering these
processes critical for understanding changes in Arctic biodiversity, livelihood, permafrost and the global carbon
budget (e.g. Zhang et al., 1996; Rennert et al., 2009; Descamps, et a., 2016; Domine et al., 2018; Serreze et al.,

99  2021).

The aim of this study is, therefore, to understand why decisions made by the snow modelling community over
the past decades have led to little or no progress in the representation of Arctic snowpack processes, i.e. in the
part of the planet that warms faster than anywhere else. While a systematic literature review would provide
some answers, this study takes a different approach borrowed from Science and Technology Studies (STS), an
interdisciplinary field whereby the scientists themselves are part of the investigation into understanding science
in the making. Although the type of decisions needed throughout the different stages of model construction has
been well documented by epistemologists and philosophers of climate science (e.g. Winsberg, 1999;
Gramelsberger, 2011; Gramelsberger and Mansnerus, 2012; Parker and Winsberg 2018; Morrison, 2021), what
leads to these decisions remains "*mostly hidden from view*" (Winsberg, 2012). Therefore, to address our aim, we
will investigate the construction of snow models by employing qualitative research methodologies, i.e. by
interviewing the individuals who shape the content of snow models in order to uncover the factors that influence
their decisions. The underlying premise of this aim is rooted in the belief that comprehending the cause of a
problem – if indeed the absence of an Arctic snowpack is one – provides a foundation for addressing it and
recommending ways to move forward, which we will do in the Discussion section

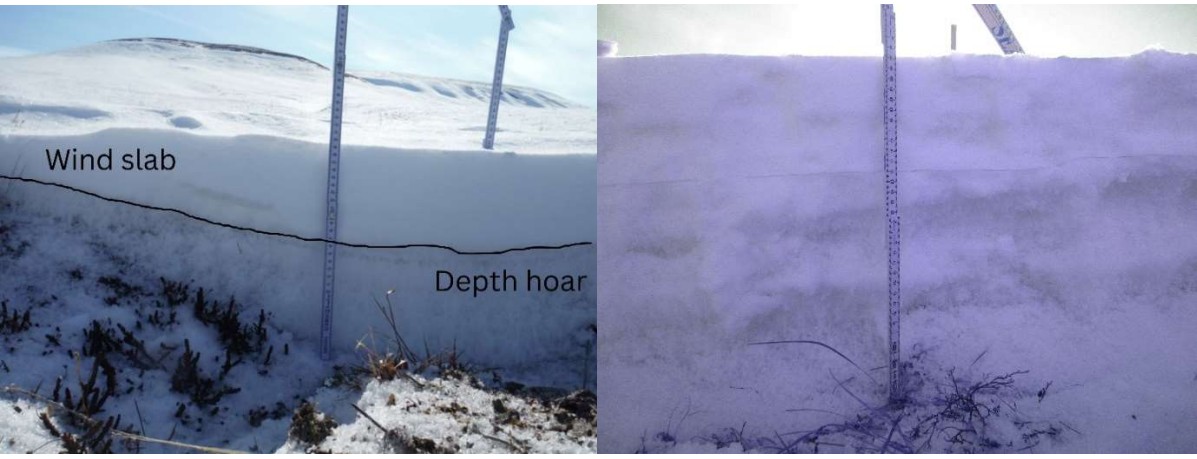

Fig 2. High Arctic snowpack with wind slab over depth hoar, taken on Bylot island on 18 May 2015 by Florent Dominé (left) and near-infrared picture showing a 2 mm ice layer at 26 cm on 16 March 2018 (right). The ice layer on the right was the result of rain on snow on 15 January. Taken at Trail Valley Creek, Canada, by Nick Rutter.

## 2    Methods

This study originated from discussions between the first three authors of this paper (CM, SR, and IM respectively) during which the representations, shortfalls and progress in snowpack modelling were debated. Our understanding was that current snow models fell short of representing all the Arctic snowpack processes needed by our project collaborators on the interdisciplinary project CHARTER, which aims at enhancing the adaptive capacity of Arctic communities to climatic and biodiversity changes (CHARTER, 2023).  For example, for reindeer husbandry and investigations into the Arctic food web, CHARTER partners required accurate snowpack density profiles and information on spatial distribution and hardness of ice layers formed by rain on snow events (see e.g. Laptander et al., 2024, for details). Recognising that we had had these types of conversations with other colleagues over the years, we concluded that a different approach was needed to understand why any Arctic snowpack processes were yet to be included in most snow models. We opted to use qualitative research methodologies because they "*place emphasis on seeking understanding of the meanings of human actions and experiences, and on generating accounts of their meaning from the viewpoints of those involved*" (Fossey, 2002). As such and in accordance with qualitative research participant selection methodology, we compiled a shortlist of participants, both within and outside CHARTER, "*who c[ould] best inform the research questions and enhance understanding of the phenomenon under study*" (Sargeant, 2012). The shortlist was split into five so-called "expert" groups:

1. Snow modeller collaborators (SMC). Participants with research expertise in Arctic fauna and flora biodiversity.
2. Field scientists (FS). Participants whose field campaigns focus on snow-related processes and whose field work supports the development of remote sensing and snow physics models.
3. Remote sensing scientists (RSS). Participants involved in the development of satellite products or of remote sensing models for snow.
4. Snow physics modellers (SPM). Participants who have developed and/or who are involved in the development a snow physics model.
5. Large scale modellers (LSM). Participants with expertise in ESMs, in the land surface component of ESMS, and/or in numerical weather prediction (NWP).

The shortlist initially included three participants in each of the five so-called "expert" groups. Potential participants were emailed with a request for participation that included a participant information sheet and consent form (see supplementary material); all those contacted accepted to participate. The groups were broadly split between stakeholders (SMC, FS and RSS), i.e. users of snow models whose needs may influence the development priorities in snow model, and snow modellers (SPM and LSM), here meaning those who make the decisions about which developments are prioritised in the snow models they are involved in. The expertise

classification was somewhat artificial and, as we discovered during some interviews, distinctions between groups were sometimes negligible. For example, all but LSM had extensive field experience, one FS had expertise in Arctic biodiversity, one RSS had been involved in the development of a snow physics model, one SPM had contributed to the development of a land surface model and so on. These overlaps prompted the addition of four more participants to the shortlist to ensure comprehensive representation of expertise within some of the groups.

In total, nineteen one-to-one interviews lasting between 40 and 65 minutes took place on Microsoft Teams or Zoom between August 2022 and January 2023. One SMC withdrew from the study shortly after the interview and their data are not used. All interviews, which were conducted by CM, were individual in-depth semi-structured interviews, a qualitative data collection method in which a set of predetermined open-ended questions, as well as themes emerging from the dialogue between interviewer and participants, are discussed (DiCicco-Bloom and Crabtree, 2006).

The description of Arctic snowpack processes and of their effects on various aspects of the Earth System was kept intentionally short in the introduction section of this paper. Implicit within the rationale for this study, is the assumption that opinions about the importance of including Arctic snowpack characteristics in snow models differ otherwise it would be no topic for debate within the Arctic snow community (here meaning all disciplines where Arctic snow is significant, thus encompassing all of this study's participants). As all participants were asked to explain the significance of snowpack structure in their research and to articulate their understanding of the importance of representing Arctic snowpacks in snow models, the implications of Arctic snowpack processes not being represented are presented, throughout the paper, in the participants' own words.

Some questions asked by CM differed between groups to reflect the expertise of the participants. SMC, FS, and RSS were interviewed to understand the diverse applications of Arctic snow (e.g. snow as a habitat, snow as an insulating medium, snow as water resource, snow as a complex microstructure etc) and to evaluate if limitations in snow models constrained their research. Interviews with individual group members followed in sequence (i.e. group 3 after 2 after 1 etc) so that SMC, FS and RSS could suggest questions to SPM and LSM. SPM and LSM were then asked about their decision-making process e.g. how do they prioritise model developments? What are the limitations of their model and how do they affect our understanding of Arctic snow processes?

All interviews were video recorded and transcribed. The data (i.e. the interview transcripts) were analysed by conducting a thematic analysis (Braun and Clark, 2006; Rapley, 2011). This qualitative analytical approach consists in identifying codes, i.e. semantic content or latent features in interviews, and then collating them into overarching themes. In our study, one or multiple codes were attributed by CM to each statement in the transcripts. Iterative coding was conducted in NVivo, a qualitative data analysis software that facilitates the classification and visualisation of unstructured data. Three iterations were necessary to identify all codes and to classify codes into themes. Codes had to be identified in multiple conversations in order to be included in the final themes. Each theme is analysed separately in the Findings sections and provided the heading of each third level subsection (i.e. 3.x.x.). The quotes that best illustrated the themes are the ones included in the manuscript and are used throughout the paper. For readability (1) speech dysfluency in quotes was edited (2) the group of the participant who is quoted is indicated before or after the quote, generally between square brackets.

Qualitative researchers must declare "*the position they adopt about a research task and its social and political context*" (Holmes, 2020) because it influences both how research is conducted and evaluated (Rowe, 2014). "Positionality" statements are necessary in qualitative research because one of the purposes they serve is to establish whether the researchers undertaking the study are "insiders" or "outsiders" to the culture under investigation (Holmes, 2020). As qualitative methods were employed to comprehend decision-making processes within a quantitative field, the positions of CM, SR, and IM as either insiders or outsiders in relation to the expertise of the participants is presented here: CM has been a model developer on snow physics and large scale models. SR and IM have been users of snow physics models. All have conducted winter and summer field work in the Arctic. All have collaborated or currently collaborate closely with all groups represented.

Finally, as was stated on the consent form signed by the participants before each interview, all participants were invited to be co-authors on this paper. This practice is becoming increasingly customary in qualitative research because it recognises that participants are joint contributors to the findings of a research project (Given, 2008; Pope, 2020). All but two accepted the invitation.

## 3    Findings: Separating the content from context

By opting for the semi-structured interview format, our aim was to use a medium, the conversation, in which using "I" was natural. The working title of this study in the participant information sheet was "*A multi-perspective approach to snow model developments*", thus implicitly alluding to the fact that, by approaching a single issue from multiple angles, this study sought to elicit diverse responses. This certainly turned out to be the case. All participants provided important information related to their field – information that is presented in subsections 3.1.$n$ –, but they also ventured where few scientists do, at least in their publications: they offered opinions. No opinion was unanimous; in fact, every statement made by each participant was contradicted by a statement made by another participant. As such, none of the quotes are endorsed by all authors and, by extension, it is expected that readers will also inevitably disagree with some quotes.

Some opinions were offered cautiously and reflected the participants' professional expertise. Others were more personal: "*I'm sick of modelers who think the world is a computer screen*", "*the scientific community is very conservative, so as soon as you try to change the paradigm, you have outcry and everyone hits each other*", "*The[se] models spend so much time doing things that aren't very important that for lots of applications, they're kind of worthless*", "*other groups have said we're going to start over, and that is also totally fraught*". Such open and candid comments do not (usually) make it to publications, but we argue that such statements are a manifestation of the participants' research identity, a concept examined extensively in education studies (e.g. Valimää, 1998; Clegg, 2008; Fitzmaurice, 2013; Borlaug et al. 2023), defined by McCune (2019) as "*the dynamic interplay over time of personal narratives, values and processes of identification with diverse groups and communities*". These processes of identification are clear in the participants' choice of words which echo McCune's definition: the participant who qualifies the scientific "community" as conservative, distances themselves from this community, as does the other one from "groups" whose strategy they reject.

The participants' research identity also manifested itself in their interpretation of the Arctic under discussion. There are many definitions of Arctic, some of which are based on the Arctic circle, treeline, climate, permafrost and so on (ACIA, 2005). CM began each interview by describing Arctic snowpack processes absent in existing models, but did not define "Arctic" beyond land snow processes, causing varied interpretations. SMC, FS and RSS, all of whom had extensive field experience, generally defined the type of Arctic they meant when describing a process, even if their description was at times itself open to interpretation: "*proper Arctic*", "*entire Arctic*", "*high Arctic*", "*Canadian Arctic*", "*tundra*", "*sub and low Arctic*", "*Scandinavian Arctic*", "*polar snowpack*", "*Finnish snowpack but not high Arctic*", "*pan Arctic*". Only two SPM and one LSM (out of four in each group) specified what Arctic they meant. No retrospective definition is provided because, despite these different interpretations, all participants knew of processes that snow models could not represent in "their" Arctic. Examples include rain-on-snow-induced ice layers, which predominantly occur in Fennoscandian oroarctic tundra, or internal snowpack thermal gradients and vapour fluxes, which are more relevant in the high Arctic.

In Section 3.1, we will outline the scientific reasons given by the participants for the lack of development of an Arctic snowpack based on the content of the interviews. In Section 3.2 we will examine the statements that deal with the context in which the participants' research is undertaken. By content we refer to the actual information being communicated, while context refers to the circumstances that help interpreting that content.

### 3.1    Content

This section presents the participants' reflections on the scientific reasons why few snow model developments have accounted for properties relevant to Arctic snow.

#### 3.1.1    Scale, heterogeneity and internal consistency

The most often cited challenges impeding the implementation of an Arctic snowpack in large scale models were related to scale, sub-grid heterogeneity and the interplay of processes within the models. The difficulty in reconciling this triad when prioritizing model developments was captured by one participant: *"[large scale models] try to represent all land processes that are relevant to all around the world for all different problems and snow, of course, is just one of however many processes that we need to be considering"* [LSM]. Therefore, *"by necessity, you have to make some trade-offs"* [FS].

These "*trade-offs*" vary in nature. One trade-off is to rank errors according to the perceived importance of the missing process as per this example: *"the spatial variability of snow depth is so high that with respect to the energy exchange with the soil below, the error that you make if you get your snow depths wrong by a few centimetres is much larger than if you miss an ice layer"* [SPM]. Another trade-off aims to maintain internal consistency in terms of complexity between the modelled processes: "W*hy would I have the perfect snow model and, at the same time, I would simplify clouds?(...) I want the model to be of the same degree of complexity in*

all its domains" [LSM]. Related to this is the opinion that "*it is undesirable in global models to have regionally specific parameterizations*" [SPM], as the inclusion of Arctic-specific processes was seen to be by some participants. This argument was countered by others who argued that, in models, solving the Arctic snowpack was not a geographical issue but a physical one: "*the physics doesn't care where it is. [Getting the physics right] should make the model work wherever*" [FS]. Finally, the last identified trade-off, which all LSM mentioned, is error compensation. Sometimes modellers know that a parameter "*is completely wrong, but it helps compensate an error in [another process. So] you have that resistance against improving a parametrization because you know that you have the error compensation*" [LSM]. For instance, for this LSM, "*in the final stages of model tuning for CMIP, I realized that error compensations had been broken away by improving the snow albedo. (…) So we [backtracked and decided not to] simulate snow albedo over the Antarctic. [We set it to] 0.77 full stop; it's completely wrong but it helped compensate an error in the downwelling long wave*".

Issues of scale are further complicated by the fact that some models are being repurposed and operate at scales that they were not intended to. Examples include models initially developed for context-specific usage now being applied globally ("*a lot of snow models are being used now in land surface schemes as broadly applicable snow models for all snow climate classes. But, I mean Crocus, it's an avalanche model, right?*" [RSS]) and large scale models increasing their resolution even though "*the physics may not be anymore realistic. It's just a little sexier to be able to say you can run an earth system model globally at 25 kilometers compared to what you used to run so*" [RSS]. Although increasing resolution means that "*processes that were before negligible are not so much so now*" [LSM], LSM ranked improving the representation of albedo or of sub-grid heterogeneity due to shading and orography was higher in the priority list than e.g. vapour fluxes.

### 3.1.2    Data availability

Model developments are supported by and evaluated against observations: "*Everything always starts at field site level in terms of testing a new model parameterization*" [LSM]. Participants from all groups (which isn't to say all participants) mentioned that more data were needed to understand the processes typical of an Arctic snowpack formation before being able to implement them in a model: "*we need to be out there when it's really happening*", "*we have very few sites across the Arctic*" so "*it's not easy with the available data. We're looking to the observations people to provide the information on the Arctic snow*" [SPM, RSS, SPM].
While the scale at which the models of the participants operate differed, all but one participant identified data gaps as being a limit to model developments. "*If you don't have site data to attribute a process to, it is difficult to defend its implementation. For example, I'm not aware of sites that we could use to tackle wind compaction*" [LSM]. Other participants highlighted the difficulty in parametrizing ice layer formation: "w*hen you find an ice crust in the snow pit, you don't know whether it is from rain on snow or wind compaction*" so "*for starters, you need the precipitation to be right*" [RSS, LSM]. While some snow physics models attempt to simulate depth hoar formation (e.g. Crocus in Vionnet et al., 2012; SnowModel in Liston and Elder, 2006; SNOWPACK in Jafari et al. 2020), data against which to evaluate the thermal gradients and vapour transport that contribute to depth hoar formation are limited; to the authors' knowledge only one such dataset, which provides both driving

and evaluation data, at a single site exists (Domine et al, 2021 at Bylot Island, Canada). However, "*it's a pretty high bar before something changes in* [large scale models] *based on a bit of experimental work. So, just because we get to show it at one site, that's not going to be good enough. You've got to show it over multiple sites, multiple regions*" [FS].

However, there is one area where snow physics models were judged to be lagging behind data availability. Five participants mentioned that the Micro-CT (Heggli et al., 2011), which allows measurements of the 3-D snowpack architecture, was a "*step-change*" [RSS] in understanding internal snowpack properties. "*Model[s are trying] to catch up with [the available data] because they now have something which is higher resolution and more objective than people looking through the microscopes handle lenses and trying to measure snow crystals on the grid, which was hugely subjective to compare to*" [RSS].

### 3.1.3   The historical development of snow models

Ten participants began the interview by providing some background about snow model developments, using this as a historical justification for Arctic snowpack properties not being included in snow models. For "*the first 30 years,* [snow physics models were] *driven by climate system processes and hydrology, snow for water resources applications*" or "*were designed to understand and predict avalanches*" [SPM, FS]. As for large scale models "*what [they] want to know about polar climate is when it influences where people live. There are people living, of course, in the high latitude, but most of the people live in the mid latitudes*" so "*every parameterization in every [large scale] model was developed for mid latitudes. And some of them work in the Arctic and some of them don't*" [LSM, LSM]. The historical legacy of model development impedes the implementation of Arctic-specific processes because the stratigraphy used in the Anderson-Kojima scheme makes it numerically challenging to adapt existing models. "*[Models] are limiting the number of [snow] layers for computational stability and efficiency so they are not respecting the way in which the snow pack is actually built up i.e. in episodic snowfall events, which will form different layers (...) That structure couldn't represent ice layers; it would refreeze meltwater or rain on snow, but in layers that are thicker than you'd observe. With numerical diffusion, these layers would spread out so there won't be a strong density contrast*" [SPM]. "*Numerically, it's just messy [to simulate the formation of an ice layer] because all of a sudden you have a new layer in the middle of other layers*" [SPM].

### 3.2   Context

This section draws on the arguments and opinions provided by the participants in Section 3.1., but with a focus on understanding the factors that influence them. Here, the arguments and opinions are framed  within the context within which the participants evolve and which the participants either implied or explicitly mentioned, As such, this section relates more to the research environment than to the science itself.

### 3.2.1 The scale of needed resources

With the exception of error compensation, which is a numerical exercise, the trade-offs discussed in Section 3.1.1 are only necessary because developments perceived to be most important needed to be prioritized. Prioritisation itself is only necessary because human, financial and computational resources are limited: "*when I speak to large scale modellers about rain on snow, the feedback is usually 'we are aware that something needs to be done, but we have other priorities and we don't have resources for this'. It's not straightforward*" [RSS].

The "*few people called 'academic scientists' [are but] a tiny group among the armies of people who do science*" (Latour, 1979). These "armies" include stakeholders, governmental research agencies, funders, taxpayers, and others, all capable of influencing funding decisions. While participants generally accepted the competitive nature of funding stoically ("*We've had trouble getting funding to do the work*", [but] "*really good and important science will not always be funded because there's not enough money to go around*" [SPM, SPM]), participants from all groups voiced concerns about the inadequate resources allocated to modelling centres given the high expectations placed on them: "*we have two groups running two different land surface schemes within the same government department on a small budget. That makes no sense*", "*that just means we're distributing our resources way too thin. Every group is tasked with doing everything - and there's a huge number of things to do in land modelling. (…) I don't think we're that far off from having a crisis situation. These models desperately need to be modernized*" [RSS, LSM]. National modelling capabilities "*need a lot more software engineering support to be able to rebuild these models, make them sleek and flexible enough that we actually have the ability to make changes more quickly without causing bugs*" [LSM].

Short-termism was also perceived to hinder progress. "*It's very difficult to make [an Arctic snowpack] model and there are also very few measurements detailing the complexity of the stratigraphy. (...) It's a long term task and it needs interdisciplinary working*" [FS]. Some participants believed that their governmental or institutional strategies impeded progress: "*[This government agency] has lots of short term goals. 'I need results for this project in six months'. Developing new tools is not part of the strategy*" [FS]. In addition, there was a recognition that short-term funding meant that modelling groups had to rely on cheaper labour in short-term employment, such as PhD students and junior postdoctoral researchers. For some participant, this meant that the type of scientific expertise required for model developments could not be met: "*You need that longevity of funding within one area. I mean, the idea that you're going to create an arctic snow model in a PhD is* [follows a non-verbal expression interpreted by CM as "mindboggling"]" [SPM]. For others, the short-termism of precarious employment impeded continuity in model building: "*you get a PhD student, (...) [they] do great work, (...) then [they're] done and [they] go on to a postdoc somewhere else*" [RSS]. The value of what is considered long-term project funding (5 years) was highlighted by an SPM: "*[this model development] would not be possible with a two to three years project. Even in five years we won't be finished, but it's still long enough to investigate the problem (...) [and to] trigger some collaborations. We are building [collaborations] between labs which will stay for longer [than our project]*".

Limited resources are also the reason why data are not available although they are not the only reason. Most Arctic research is conducted by researchers who are not based in the Arctic, which is a logistical reason why "*the number of detailed measurements in the Arctic during the entire winter season is close to 0*" [FS]. "*If you

*want to study alpine snow* [e.g Col de Porte, France, and Davos, Switzerland, which were set-up to support the
local tourism industry], *you get out of your home, walk in the field or take your car, drive 15 minutes and you*
*see it. If you want to look at arctic snow, it's more complex"* [FS]. The nature of this complexity is manifold.
Firstly, although no participant mentioned that meteorological instruments are prone to malfunctioning at low
temperatures (see e.g. Fig. 3), it was understood to be the implicit reason why some measurements were not
available. Secondly, *"we need to find people willing to do this work in total darkness"* [FS]; polar nights and
harsh winter meteorological conditions make access to Arctic sites difficult, which is why field campaigns often
take place in Spring and Summer time. However, *"we need to observe how this happens in the real world. I*
*mean, we certainly have snow pits and we see ice lenses there, but we need to be out there when it's really*
*happening"* [SPM].

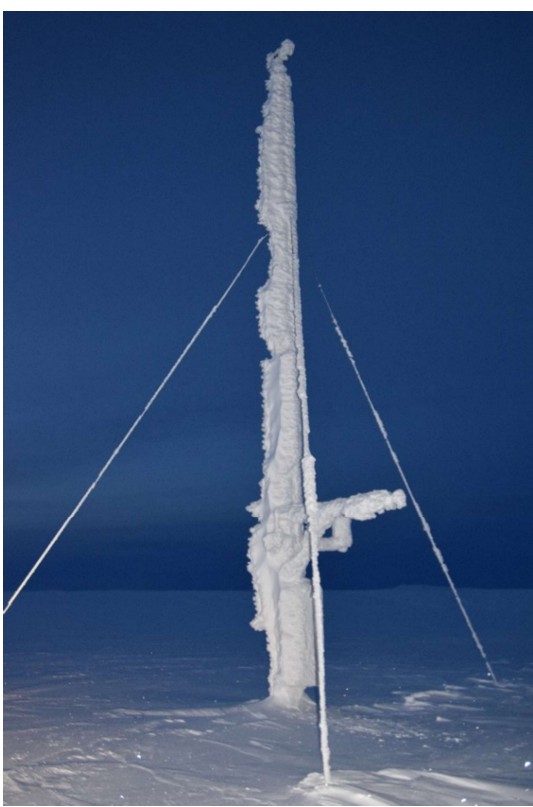


*Figure 3: Meteorological station covered with rime before maintenance in Reinhauger, Varanger peninsula, Norway. Photo*
*taken on 23 January 2020 by Jan Erik Knutsen.*

**3.2.2    Adaptability**

Public funding is granted to projects that fall within the strategic objectives and research priorities of
government funding agencies. As such, *"the right to research"* (Henkel, 2005) is conditional upon scientists
adapting and responding to an evolving funding landscape. Although much literature argues that there is a
conflict between academic freedom and solution-based or applied science (e.g. Henkel, 2005; Winter, 2009;
Skea, 2019 etc), we found instead that adaptability and shifting priorities was integral to the modellers' research
identities. *"To some degree, we follow what is being hyped, you know, if something is being hyped in Nature"*

[LSM]. Model developments were presented as being responsive and at the service of others: "*There is no master plan. It's opportunity driven, it depends on projects that come in, (...) on what some of the users want to do. It's kind of nice*" [SPM]. When questioned about what the priorities for snow model developments are, one SPM answered "*It's not just the snow modellers who can answer that. It is the people who want to use the snow models*". Arguably, performance-based research funding systems like the UK Research Excellence Framework have been in place long enough in some countries for researchers to have adapted to the constraints of the "publish or perish", "be funded or fade out" and "impact or pack in" culture.

In fact, interdisciplinary collaborations were the key motivation for model development, demonstrating the modellers' adaptability. The reasons for interdisciplinary collaborations driving snow model developments were manifold. First, they are necessary to address research questions: "*Permafrost, snow, wildlife biology (...) These fields have evolved independently over the last 30 or 40 years or whatever (...) [Now] we're working together to do a better job of answering all these interdisciplinary questions*" [SPM]. Second, they drive innovations in all fields involved: "*if you don't have a good physical snow modelling capability, you can't maximize the value of new [satellite data] retrieval algorithms*" [RSS]. Third, they allow model developments to be relevant to a wide range of stakeholders, as is, for example, the case with progress on the many sectors that rely on numerical weather predictions. Fourth, they generate funding: "*We wouldn't have enough base funding to pay for a master plan [for model developments] so we are depending on projects that come in and on the interest of individual people*" [SPM].

A particularly topical illustration of the significance of interdisciplinary collaborations for snow model development at the time of the interviews was the IVORI project (IVORI, 2023), which aims to develop a new type of snow model that will be able to model the snowpack processes existing models cannot. IVORI was mentioned spontaneously by eight participants other than the project lead (herself a participant in this study). "*We had a consultation meeting at [a conference] in 2016. It was really mostly the snow community just saying, 'hey, we want something better' (...) The ice core community was also pushing in this direction (...) [as well as] the remote sensing community [because] no model correctly represents snow microstructure [they need]*" [SPM]. Although all participants were cautious not to oversell a model at a very early stage of its development, there was a lot of excitement around the project: "*[IVORI] is trying to basically rethink the whole snow modelling issue from scratch and come up with a new model that will be the future*" [SPM other than the IVORI project lead].

Finally, collaborations provide human resources, especially when models are open-source. From the developers' perspective, open-source means that "*the majority of the development work is done external[ly. For example,] for the most recent release, we had 50 people involved from 16 different institutions*" [LSM]; for the users, it makes models "*easy to use. You can just pick up examples and test the model for yourself (...)*" and "*if something doesn't work or if you have questions, you always find support*" [RSS, LSM].

### 3.2.3 The anchoring bias

Some participants in all but the SMC group argued that many developers misjudged or did not understand the importance of snow when modelling Arctic processes. Four participants stressed the need to design and to implement a long-planned snow model intercomparison project (SnowMIP) focusing on tundra (in both Arctic and Antarctic) snow processes because "*the first thing it would do is alert the modelers to the difficulties that they have in the Arctic that, in the absence of these evaluations, they wouldn't even know about… In my sense, large scale climate modellers aren't sufficiently aware of snow. (…) There are so many people who don't care about that*" [LSM].

One of the reasons whysome modelers "*wouldn't even know about (…) the difficulties that they have in the Arctic*" is because their existing models served as "anchors" or benchmarks. Anchoring is a common cognitive strategy where individuals, including experts, rely too heavily on an initial piece of information that they use to assess risk and uncertainty, leading to systematic errors in judgements (Tversky and Kahneman, 1974). Within our context, it means that even though some participants acknowledged that existing models represented Arctic snow processes poorly or not at all, the fact that they represented snow at all meant that some participants preferred not taking the risk of investing resources into new models or time-consuming complex developments.

For example, when an SPM said a "*model is never perfect, but is it good enough for what is being done with it*?", what they interpret as being "good enough" is contextual. It depends on the research question to be addressed, on the data, time and funding available, on the extent to which what is expected of the model measures against the anchor. As such, what is "good enough" is dynamic and evolves as the anchor or reference point shifts. For one LSM, the anchor shifted during the interview: "*I understand now what you* [CM] *have been talking about, how far we are from what people who live in the Arctic really care about*". This insight, along with the historical development context outlined in Section 3.1.3, suggests that the anchoring bias in snow modelling partly reflects non-epistemic values (hereafter simply referred to as values), i.e. ethical and social considerations that help scientists make decisions which do not rely on expertise alone (see e.g. Rudner, 1953; Winsberg, 2012). For instance, the historical context outlined in Section 3.1.3 echoes value judgments prevalent in early model evolution that prioritized serving the majority of people who live in the mid-latitudes.

The anchor, or benchmark against which to evaluate model priorities, also shifts as a result of community efforts such as MIPs, which motivate developments because they "*distil the information and tell [modellers] what are the priorities and what are the sites good for. (…) [SnowMIP] brings together observation experts and other models and modellers. We all learn enormously*" [LSM]; "*the community does a reasonably good job of trying to develop, incrementally, through different research groups*" [FS]. Nevertheless, as "*models are not [currently] very well tested for the Arctic, it is not easy to know what they do well*"[SPM], anchoring bias plays an important part in the assessment of whether models are "good enough" or not.

Finally, eight participants spontaneously discussed the risks and benefits of starting models from scratch in view of ongoing projects undertaking this task (e.g. CliMA, 2023, a novel type of climate model, and IVORI). While the time and effort of such an undertaking were the main causes for concern ("W*ith respect to the new model, what I see is that this quest for purity (…) makes things extremely slow"; "the effort of rewriting a climate model [is huge]. I'm not saying it's not worth it (…) but I can understand why people don't do it*" [SPM, LSM]), it is because the participants were weighing the value of starting from scratch against, instead, a reference or

anchor point i.e. the existing models, that one concluded that starting from scratch was *"totally fraught because you're probably talking about a five year project to get even close to the capability of what the current models have. And at the moment, who wants to give up their capabilities?"* [LSM]. On the other side of the argument, a FS argued that "*trying to improve the candle did not invent electricity. [For tundra snow], existing snow models, there's one thing to do with them. Trash*". Somewhere in the middle, more nuanced opinions were presented: "*The community should be endorsing IVORI, but there is such a lag between activities like this and the current suite of models, which people use in high impact papers, that we also need to spend time understanding what the limitations are and how we can get some improvement out of these models*" [RSS].

## 4 Discussion and moving forward

As mentioned in the Introduction, the premise of this study was rooted in the belief that comprehending the cause of a problem would provide a foundation to address it. The premise found echoes in this RSS's quote:*"[You] should never keep doing what you're doing because that's the way it's always been done. (...) What are the priorities? What do we need to learn? What do we need to do that's new?"*. Sections 3.1. and 3.2 showed conflicting answers, opinions and perspectives to these three questions. In this Discussion section, we aim not to reconcile these opinions, but, based on our reflections of the findings, we will aim to start answering these questions to propose ways forward.

### 4.1 Opening-up research

As mentioned in Section 3.2.3, values have contributed to deciding priorities for snow models development over time, such as the importance attributed to their relevance to where "*most of the people live*" [LSM] e.g for their survival (e.g. water resources) or leisure (e.g. avalanche forecasting). As mentioned in Section 2, SMC, FS, and RSS were interviewed to provide a broad picture of the range of Arctic snow applications and to understand how the absence of an Arctic snow model constrained their own research. Because of the different role that the Arctic snowpack plays in their research, these participants reframed snow models away from their historical model legacies into research areas seen as being underexplored by the Arctic snow community. They proposed how efforts to represent Arctic snowpack processes could pave the way for new interdisciplinary collaborations, yielding benefits such as innovation, stakeholder involvement and funding:

Permafrost-carbon feedback "*Snow is a kind of blind spot in the international climate modelling community. We know that snow is wrong, but people are not coordinated, people are not really working together*" [LSM]. "*At the moment, snow structure is not considered for permafrost modelling. It's only how thick the snow is and whether the temperature decouples from the ground or not*" [RSS]. Participants from all groups highlighted the importance of snowpack structure to understand soil winter processes. "*It's clear that the winter climate is changing even more than the summer climate*" [SMC]. For example, "*when there is rain-on-snow, the short-term warming to the ground influences the entire following winter history. What is the magnitude of the impact? Knowing the temperature at the base of the snow is the really crucial information*" [RSS]. One participant

stressed the importance of upscaling the many *in situ* soil experiments with the help of suitable snow models: "*What manipulation experiments show is that whether we have less snow, or shorter winters or we have ice layers or something else will have very different, even opposite, effects on soil processes, gas exchanges, plant and soil ecology. (...) For example, when you have ice layers, the ice is disturbing the gas exchange between the soil and atmosphere, but it's still active (...) [so] you get carbon dioxide accumulation. We also found that soil microbes are resilient to late snowpack formation and earlier melt, but the growing season started earlier than usual. (...) [What we now need] is to translate the results of that experiment to larger landscape level*" [SMC]

Arctic food webs Upscaling is also needed to translate local scale findings to ecosystem scale when investigating fauna biodiversity. "*When the snow gets very hard [e.g. after a ROS event or refreezing], lemmings don't move as well through the snow; they cannot access their food anymore and then they starve (...) [Many] specialized Arctic predators depend on lemmings to survive (...) or to reproduce successfully [e.g. snowy owls, pomarine skuas, Arctic foxes]. (...) They also eat a lot and influence the vegetation (...) If a snow model could reconstitute the snowpack in a reliable way, we could see if there a relationship at the large scale between cyclic lemming populations and snow conditions? (...) and address a row of other ecological hypotheses*" [SMC].

Reindeer husbandry For reindeer herders, obtaining near real time spatial information on the structure of the snowpack could save their livelihood and their lifestyle: "*During the winters of 2020 and 2021, we had thawing, raining and refreezing in January and there was already a lot of moisture at the ground from the previous Fall. So the reindeer have to dig through all that and then there's a layer of ice on the ground. The lichens, blueberries, everything is encased in ice. So there's two options. They starve or they short circuit their digestive system because they eat the ice-encrusted vegetation get too much of water in their rumen. The Sami herders say that kills the animal anyway. (...) If the herders could get a heads up (...) Can I go move my herd? East. West. Where is soft snow?*" [SMC].

Remote sensing applications Remote sensing products are used to tackle many environmental issues, including the three described in this section and their development is intrinsically linked with physically-based models. "*Remote sensing doesn't work everywhere all the time so we need to combine information from a model and from satellite data. We need to improve the physical snow models, but in step with developing the remote sensing. If you do one without the other then you're not gonna be able to maximize the value of both*" [RSS]. For example, "*snow has a confusing effect on retrieval estimates. Some of the signal comes from the atmosphere [e.g. clouds], some comes from the snow, and if you can't disentangle what comes from what then you just throw away millions of satellite data that could potentially be used for numerical weather prediction, better weather forecasts*" [RSS].

**4.2    A plurality of strategies**

Discussions about trade-offs in model building (as in Section 3.1.1) precede the development of the first general circulation models (Manabe, 1969), the core components of ESMs, which already included snow. In 1966, Levins argued that, given computational constraints that remain valid six decades later, models could not be

general, precise and realistic at the same time; when designing their model building strategy, modellers had to
choose which property to trade off. Levins concluded that as no single model strategy could represent a complex
system, a plurality of models and model strategies was necessary to provide a more comprehensive picture of
the system.
The different opinions expressed throughout this paper suggest that the participants support different strategies.
The strategies they endorse are partly dictated by different local epistemologies, i.e. assumptions, methodologies
and aims specific to a community (Longino, 2002), and disciplinary identities, i.e. discipline-specific socio-
historical norms (Dressen-Hammouda, 2008).  For example, ESMs must sacrifice realism and so must, by
extension, LSM: ESMs are precise because they use equations that provide precise outputs, general because
these equations must be applicable globally, but have unrealistic internal processes (e.g. see error compensation
in Section 3.1.1). However, within groups disagreements and between groups agreements also show that
disciplinary identity and local epistemologies do not always dominate the research identity narrative of the
participants. As noted in Sections 3.2.2 and 4.1, collaborations are drivers for model developments and, when
interdisciplinary, these collaborations will also shape the research identities by exposing them to different
disciplinary identities and local epistemologies. For example, as mentioned in Section 3, one FS declared
that they are "*sick of modelers who think the world is a computer screen (...). If you haven't been in the field (...)*,
*you just don't understand what's going on*". However,  another FS declared that "*there are people doing*
*fantastic snow modelling work who don't really see a lot of snow, but they've got the appreciation of*
*understanding what the detail is. It helps to see [on the field] what you're looking at [on your screen], but it's*
*not an absolute essential*". The two FS manifest clear differences in their value judgments, with the first one
valuing empirical evidence and lived experience over theoretical knowledge and the second having "*become a*
*bit more nuanced in [their] thinking*" after having been "*exposed to different types of models*". Historically, the
notion linking value-free science with objectivity and impartiality has prevailed (Pulkkinen et al., 2021) and was
an obstacle to bridging the gap between our personal identity, reflected in our values, and our research identity,
reflected in our professional decisions (Staddon, 2017).  However, the role that non-epistemic values play in
climate science was recognised in a dedicated subsection (1.2.3.2) of the IPCC WG1 AR6 (IPCC, 2021), thus
providing a space for these conversations to occur in a field historically dominated by epistemic values (e.g.
truth, accuracy, falsifiability, replicability).
While Levins' plurality of model strategies was originally aimed at model building in population biology, its
relevance has been extended to climate science by, amongst others, Parker (2011), Lloyd (2015), Morrison
(2021), Walmsley (2021) and Winsberg (2021). They argue that diversity of opinions, values, epistemic
pluralities and strategies do not need to be resolved, but, on the contrary, that a plurality of models that
investigate the same phenomenon from different representational perspectives is necessary. One of the most
prominent examples in which climate science exploits this plurality is via MIPs, which aim to assess "*the*
*robustness, reproducibility, and uncertainty attributable to model internal structure and processes variability*"
(IPCC, 2021).
However, considered together, existing snow models do not provide this plurality of representational
perspectives necessary to understand a complex system. Instead, many of these models are interdependent
(Essery et al., 2012) and, rather, provide a plurality of representational complexities all based on the same
Anderson-Kojima scheme. Aligning with epistemologists of science, we argue that developing a snow model
adapted to Arctic snowpack processes to complement existing models is, therefore, arguably necessary to
provide the diversity in model strategies needed to understand complex Arctic processes and interactions.

**4.3    Snow model intercomparison projects**

The Earth System Modelling – SnowMIP (ESM-SnowMIP; Krinner et al, 2018), the fourth snow model
intercomparison in 24 years (Slater et al, 2001; Etchevers et, 2004; Essery et al., 2004; Rutter et al, 2009; Essery
et al., 2009) is a community effort that aims to evaluate snow schemes in ESMs and to improve our
understanding of snow-related feedback in the Earth System. Out of the ten planned exercises, only two have
taken place so far (Menard et al, 2021; Essery et al., 2021). During the first exercise, little progress in snow
models was found to have occurred since the previous snow MIPs (Menard et al., 2021) because of scientific
reasons as well as contextual circumstances that resonate with the findings in this study.
The next planned phase, which aims to test models in the tundra, has suffered a number of setbacks, not least
because *"the models are not very well tested for the Arctic so it is not easy to know what they do well and it's*
*not easy to ask that question with the available data"* [SPM]. In line with discussions about responsible
modelling in other sectors (e.g. Saltelli et al., 2020; Nabavi, 2022), we argue that by involving stakeholders (e.g
as represented here by SMC, FS and RSS) in future snow MIPs, the models would be better prepared to tackle
research questions that currently remain unanswered (although there have been attempts to do so with the
existing models), thereby unlocking opportunities in new research domains and motivate the collection of the
new type of data needed to test models in the Arctic (Sections 3.1.2 and 3.2.1). The research questions identified
in Section 4.1 should contribute to determining the focus of the next snow MIP rather than the next snow MIP
determining what questions can be answered given the current modelling constraints, the latter approach failing
to challenge the notion that existing models are "good enough".
Another consideration would be what legacy a tundra SnowMIP would want to leave behind. In the past,
SnowMIP participants were required to provide model results. However, if a tundra SnowMIP is to advance
snow modelling, the obstacles that limit the implementation of Artic tundra snow processes (see subsections
**Error! Reference source not found.**.x) should be directly addressed. One suggestion mentioned by
participants, although not within a SnowMIP context, was that moving forward, *"shareable modules would be*
*strategies that would allow us to make better progress"* because *"it will be easier for people to take your*
*parameterization, take your model compartment and put it in their model to see what it does"*. We argue that
future snow MIPs should be vehicles to foster more direct collaborations between modelling teams and with
users by advocating for sharing of, amongst others, code, results and configuration files. This would avoid
duplication of efforts and accelerate the model developments required to tackle Arctic snow challenges.
However, *"a modelling centre doesn't get money to do a MIP, but they want to do it because it's important to*
*them. So, they end up being involved, but they get MIP-saturated and that's when the errors arise (…) At the*
*very least, future SnowMIP-like projects need dedicated people whose main responsibility is to take this on, to*
*say 'I have funding to do it, I can dedicate time to it'"* [RSS]. Lack of funding towards MIPs participation is one
of the many contextual factors Menard et al. (2021) identified as hindering the first ESM-SnowMIP exercise.
Unless the context in which MIPs, SnowMIP or otherwise, operate is reconsidered, the same factors will
continue hindering community efforts.

### 4.4 Modeller accountability and empowerment


Models are not only the representation of a situation, but also the product of many socio-political interactions
(Nabavi, 2022). Even when models lack core government funding, the ability of modellers (as defined here in
Section 2) to secure competitive funding underscores their alignment with strategic research priorities that often
reflect political agendas. Heymann and Dahan Dalmedico (2019) argued that the IPCC ushered in a new era of
expertise in which scientists are conditioned and formalized by politically relevant issues; as architects of ESMs,
this implies that modellers become vehicles for political agendas.
Participants in this study have provided various reasons for not having prioritised the development of an Arctic
snowpack model: data availability, historical context, human resources, lack of funding, competing research
priorities, strategic priorities of government agencies and so on. In Section 4.2, we discussed the role of values,
which are situated within a social and political context, in these decisions. We argue that they warrant more
transparency in revealing the position of modellers within these contexts. We suggest that, following Bourdieu
(2001) who argued that scientists should not take a position without acknowledging that they are doing so,
natural scientists should position themselves as "insiders" and "outsiders" within the context of the research they
conduct and publish (CM, SR and IM followed this advice themselves in Section 2). "Coming clean" (Lincoln,
1995) about our positionality in our publications would foster a more responsible research environment and
contribute to the ongoing discussion about the role of values in climate science, as explored Section 4.2. For
instance, weaknesses in the reviewing process as described below may be avoided if positionality statements
allowed journal editors to identify gaps in the authors' expertise: "*Some papers (...) don't make the effort to*
*quantify what the sensitivity of their key result is to how snow is characterized in the model. [For example, if the*
*paper is] (...) about carbon budgets across the Arctic for over 12 months seasonal cycle, [the review] always*
*goes towards the growing season community (...).So [these papers] don't get scrutinized the way they should"*
[RSS].
Finally, a "*unique practice of sensitive wording*" (Gramelsberger et al., 2020) was developed in climate science
to describe the information produced by climate models. This practice satisfies the socio-political expectations
of climate science to produce trusted information in decision-making, as well as acting as a barrier to accidental
or intentional misinterpretation of the same information by climate deniers. An example of such sensitive
wording is the "likelihood language" used to describe scientific uncertainties (Landström, 2017; Moss &
Schneider, 2000). We suggest that another instance of sensitive wording is the separation between the model
and the modeller, which contributes to presenting the information produced as objective and impartial. For
example, the IPCC WG1 AR6 mentions "*model(s)*" 12666 times, but "*modeller(s)*" three times. Such wording is
invisibilising the role of modellers in the decision-making process of model development and evaluation, and
arguably, in some of the information produced in climate science. Yet, models are a product of one or multiple
modelers' vision. This was reflected in the interviews during which more participants referred to Richard's
model, Glen's model or Marie's model rather than to FSM, SnowModel and IVORI respectively. David
Lawrence was named by all participants who mentioned CLM, as was Michael Lehning for SNOWPACK.
Crocus was the only model that a large majority of participants did not associate with any particular modeller.
The research identity of many modellers is, whether they want it or not, intertwined with their model; inviting
authors to reflect about their positionality would allow them to regain control over their own narrative and
research identity.

**5    Conclusion**

As per more conventional review papers, the novelty in this paper is not in its content, but in the medium it
chooses to present that content. What participants said, they had said, but not necessarily written, before.
Conferences, workshops, meetings and end-of-day visits to more informal venues are places where
disagreements about the limits and motivations to model development *are* debated. But while the written history
narrated by our publications does record the arguments presented here in the content section, it does not record
what is presented in the context section.
In fact, the medium is not novel either. Science and technology studies examine the context within which
science is constructed and philosophers of science have long debated the decision-making process of scientists.
As such, much of what is non-Arctic snowpack-specific could probably be found in many of these disciplines'
seminal texts. However, although one of the participants directly quoted one of Thomas Kuhn's, a pioneer of
STS, concepts when they advocated for a change in paradigm (Kuhn, 1962), STS is practiced by outsiders
looking in on a field, as is philosophy of science. These positions hinder the dissemination of their findings to,
and the acceptance of their recommendation by, insiders.
Therefore, the novelty here is that it is an insider's job. It is a reflective exercise which, we hope, will be the
start rather than the end point of the conversation. The comments of the participants-turned-co-authors at the
paper writing stage certainly suggested so much: "'*it's interesting that nobody commented on the conventional*
*wisdom that modelling tundra snow is "too hard"?*'; "*discussions about digital Earth twins are shaking the*
*[LSM] community. Some suggest that many resources, on continental or even global level, should be bundled to*
*create* the *one big model. Others think this is a recipe for disaster, and some that is "scientific colonialism'*";
"*the next step in modelling should be an evolutionary one: we should take the best of each*".
The participants were interviewed in their role (or identity) as researchers, but all will have been reviewers of
papers and grants, some (co-)editors of journals and some will have influenced policy-makers. We argue that it
is our role as insiders to motivate the change to our own practice. We also argue that it is our role as researchers
to be more transparent about the contextual factors that influence and restrict our decisions. More importantly, it
is our role as reviewers, editors and policy-makers to allow for such transparency to happen and to challenge
openly the idea that short-term funding can lead to ground-breaking science, that Arctic data can be collected
without engaging the people who live there, that 40-year old models are good enough to tackle challenges we
knew nothing about ten years ago. If we fail to take on these roles, the reality of our scientific process will
remain invisible and silent, and by virtue of it being hidden, unchanged.

## 6   Code / data availability

The transcripts are not available as they contain sensitive and personal information.

## 7   Author contribution

CM, SR and IM conceptualised the research. CM conducted the interviews and analysed the data. CM prepared the original draft with contributions from SR and IM. All other co-authors were interviewed for the research and contributed to the final version of the manuscript.

## 8   Competing interests

At least one of the (co-)authors is a member of the editorial board of The Cryosphere.

## 9   Acknowledgement

CM, SR and IM thank all co-authors, Michael Lehning, Juha Lemmetyinen and the one anonymous participant who withdrew from the study for being interviewed. We thank Jan Erik Knutsen for providing the photo used in Fig. 3. This project was funded by the European Union's Horizon 2020 programme (CHARTER, grant Nr. 869471). Marie Dumont has received funding from the European Research Council (ERC) under the European Union's Horizon 2020 research and innovation program (IVORI; grant no. 949516)

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
