# Peer review of "Exploring the decision-making process in model development"

_EGUsphere, 2023_

## Author Comment (AC1)

Review of "Exploring the decision-making process in model development: focus on the
Arctic snowpack" by Menard et al.

We thank the Reviewer for taking the time to review our manuscript. Please find our
answers in blue font.

This is a somewhat unusual manuscript, submitted as a "Research article" for consideration
in The Cryosphere.  The "unusual" aspect is that, where most research articles focus on
measurement data and/or simulations, this study reports on interviews conducted with
experts in the field of Arctic snow, [L96-97] *"to understand why decisions made by modellers*
*all over the world and over the past decades have not led to more (or is it "any"?) progress in*
*Arctic snowpack modelling, …"*

It is true that "*[quantitative] research articles focus on measurement data and/or*
*simulations*". However, qualitative research does not. Indeed, it is "*unusual"* for a
manuscript using qualitative methodologies to be submitted to a journal that predominantly
publishes research using quantitative methodologies. For this reason, we contacted the TC
editorial board prior to submission to check whether a manuscript using qualitative
methodologies to investigate decision-making in snow modelling could be considered for
peer-review. The editorial board confirmed that qualitative methodologies as applied to
cryosphere topics are within the remit of the journal.

We suspect that the many comments in this review questioning the methodological
soundness of our study may stem from Reviewer 1's unfamiliarity with qualitative
methodologies. We did expect that some TC readers would be unfamiliar with qualitative
research, which is why we did explain our process throughout the manuscript, but now
realise more information will be needed. For example, we described our approach in the
Methods section and referenced a number of papers examining the qualitative
methodologies we used in our manuscript (e.g. Braun and Clark, 2006; DiCicco-Bloom and
Crabtree, 2006; Lincoln 1995; Rapley, 2011). We as well mentioned in the Introduction and
Conclusion that our approach was borrowed from Science and Technology studies (L99 and
L563-568).
Nevertheless, the comments from Reviewer 1 made us realize that, in our revised version,
we will need to provide more information about our methodology and stress in greater
depth the value and complementarity of qualitative research. We will for example quote
Fossey (2002) in the introduction to set the tone of our work: "'*Restricting oneself to any*
*single paradigm or way of knowing can result in a limitation to the range of knowledge and*
*the depth of understanding that can be applied to a given problem situation'. (…) Thus,*
*research needs to draw on different perspectives, methodologies and techniques to generate*
*breadth of knowledge and depth of understanding. Qualitative research is a broad umbrella*
*term for research methodologies that describe and explain persons' experiences, behaviours,*
*interactions and social contexts without the use of statistical procedures or quantification.*
*(…) One of the major criticisms is that within the positivist paradigm* [i.e. scientific research
based on quantitate methodologies] *it is assumed that an objective reality, or truth, exists*
*independent of those undertaking the inquiry and the inquiry context. Two research*
*paradigms that inform qualitative research methodologies, namely the interpretive and*

*critical research paradigms, place emphasis on seeking understanding of the meanings of*
*human actions and experiences, and on generating accounts of their meaning from the*
*viewpoints of those involved*"

Here, issues identified are the somewhat troublesome transferability of modeling
approaches between lower latitudes and polar regions, limited data availability from the
arctic suitable for model development, parameterization development, and calibration and
validation. Other issues are the historical underrepresentation of arctic snow in snow model
development environments, lack of attention and (thus) funding for the problem, and
inadequate approaches.

Major comments:

1.  I think it is an interesting concept to access knowledge that is normally not finding its
way to the broader community in the form of manuscripts. However, I think there are
some methodological problems that devalue this manuscript from a "Research article"
to only an opinion piece.

1.  First of all, the selection of participants was seemingly done very subjectively,
  and is not transparent for the reader. The only procedural aspect mentioned
  here is [L118] *"CM, SR and IM compiled a shortlist of participants"*. I wish that
  there would have been some objective criteria, for example a random pick of
  first authors on papers that mention "snow", "arctic" and "modeling" in the
  abstract that were published over the last ten years, based on a database like
  Scopus, ISI knowledge or google scholar.

  Participant selection abided with qualitative methodologies: "*Quantitative*
  *research requires standardization of procedures and random selection of*
  *participants to remove the potential influence of external variables and ensure*
  *generalizability of results. In contrast, subject selection in qualitative research is*
  *purposeful; participants are selected who can best inform the research*
  *questions and enhance understanding of the phenomenon under study (…)*
  *Decisions regarding selection are based on the research questions, theoretical*
  *perspectives, and evidence informing the study.*" (Sargeant, 2012)

  Based on this, we explained our reasoning for selection in the introduction L96-
  104: "*The aim of this study is to understand why decisions made by modellers all*
  *over the world and over the past decades have not led to more (or is it "any"?)*
  *progress in Arctic snowpack modelling (…) Therefore, to address our aim, we will*
  *investigate the construction of snow models through interviews with the*
  *individuals who shape their content and present the results of this investigation*
  *in their own words*". Nevertheless, we recognise that TC readers may welcome
  more information about our selection process, which we will provide in the
  revised version in the Methods section.

For completeness, we note that that the quoted sentence was truncated. The
full sentence is "*CM, SR and IM compiled a shortlist of participants, both within*
*and outside CHARTER, who consider the snowpack structure important for their*
*research*".

2. Second, the manuscript relies heavily on quotes from the interviews. The full
interview transcripts are, understandably, not released. Thus this could
potentially result in heavy cherry-picking of quotes by the first three authors.
Apparently, the interview transcripts have been coded using NVivo, but it is not
clear how this has further been used. It is not clear what attempts were made
for objective analysis of the interview transcripts.

Information about our methods is found between L152:156: "*The transcripts*
*were analysed by conducting a thematic analysis (Braun and Clark, 2006;*
*Rapley, 2011), which consists in identifying codes (semantic content or latent*
*features in interviews) and collating them into overarching themes. Iterative*
*coding was conducted in NVivo, a qualitative data analysis software that*
*facilitates the classification and analysis of unstructured data. Three iterations*
*were necessary to identify all codes and to classify codes into themes*". We
recognise that the TC readership may welcome more information about
thematic analysis and will provide more information in the revised manuscript;
readers particularly interested in the methodology are invited to consult the
two papers cited above. Until then and briefly: The questions asked during the
interviews were to understand the decision-making process of the participants.
Following the interviews and using the transcripts, one or multiple codes (or
"labels") were attributed to each statement by CM. Codes were then merged
and grouped into themes. This process was repeated three times to ensure
thorough codification. These themes were then addressed separately in each
subsection in Sections 3. The quotes that best illustrated the themes were then
included in the manuscript.

3. Third, I'm concerned that quotes from the interviewed scientist are published,
without fact-checking if this is true. This results in a few false statements, for
example that *"CROCUS is an avalanche model"* [L237], or that [L282-
283] *"[Models] are limiting the number of [snow] layers for computational*
*stability and efficiency"*, which for Crocus or SNOWPACK, for example, would be
trivially easy to adjust. [L373-375] *"In my sense, large scale climate modellers*
*aren't sufficiently aware of snow. (…) There are so many people who don't care*
*about that"*. The first part of this statement is an opinion. The second part is
stated as a fact. *"There are so many people who don't care about that"*. I would
like to see evidence for that.

We had attempted to explain the nature and purpose of the quotes L173-177 in
the manuscript: "*The working title of this study in the participant information*
*sheet was "A multi-perspective approach to snow model developments", thus*
*implicitly alluding to the fact that, by approaching a single issue from multiple*
*angles, this study sought to elicit diverse responses. This certainly turned out to*

*be the case. Most significantly, no opinion was unanimous; every statement*
*made by each participant was contradicted by a statement made by another*
*participant*." We also addressed the fact that we were eliciting opinions "*By*
*opting for the semi-structured interview format, our aim was to use a medium,*
*the conversation, in which using "I" was natural. While all participants provided*
*important information related to their field – information that is presented in*
*Section 3.1– they also ventured where few scientists do, at least in their*
*publications: they offered opinions*". Reviewer 1's comment makes us realize
that the nature of the statements may not be clear to all readers and we will
therefore clarify in the next version of the manuscript that all quotes are
opinions and that as some quotes contradict each other, none of the quotes are
endorsed by all authors and, consequently, readers will inevitably disagree with
some quotes.

Regarding "fact-checking" and "truth", as explained in the manuscript and
above, what we are interested in is how the opinion of decision-makers - in
other words their truth based on their experience, expertise and perspective -
inform their decisions. We also stress that the word "fact" is loaded in social
sciences, including Science and Technology studies (we refer Reviewer 1 to
Fleck, 1935, referenced L565), which understands facts as being constructed.

Regarding the Crocus quote, we invite Reviewer 1 to read the full paragraph in
which it figures to understand the context in which it is cited: "*Issues of scale*
*are further complicated by the fact that some models are being repurposed and*
*operate at scales that they were not intended to. Examples include context-*
*specific models being used at large scale ('a lot of snow models are being used*
*now in land surface schemes as broadly applicable snow models for all snow*
*climate classes. But, I mean Crocus, it's an avalanche model, right?')*". We hope
that, as we will be more explicit about quotes expressing opinions in the revised
version, there will be no more room for misinterpretation. We will also revise
this paragraph e.g. "*some participants believed that some models were being*
*repurposed and operated at scales that they were not initially intended to*"; this
will help the reader understand that the quote means that the participant does
not think that a model initially developed as an avalanche forecasting model
(Brun et al., 1989) should not be broadly applied for all snow climate classes.

• *"[Models] are limiting the number of [snow] layers for computational stability*
*and efficiency*", which for Crocus or SNOWPACK, for example, would be trivially
easy to adjust. [L373-375].
As mentioned above L134-139, we will clarify that all quotes are opinions. We
also invite Reviewer 1 to review the context in which this quote is cited: "*Ten*
*participants began the interview by providing some background about snow*
*model developments, using this as a historical justification for Arctic snowpack*
*properties not being included in snow models*".

• *"In my sense, large scale climate modellers aren't sufficiently aware of snow. (…)*
*There are so many people who don't care about that*". The first part of this statement is an opinion. The second part is stated as a fact. *"There are so many*
*people who don't care about that"*. I would like to see evidence for that.
As mentioned above L134-139, we will clarify that all quotes are opinions.

4.  Lastly, the Interview consent form states: *"Access to the interview transcript will*
*be limited to the research team: Dr Menard, University of Edinburgh; Dr Sirpa*
*Rasmus, University of Lapland; Dr Ioanna Merkouriadi, Finnish Meteorological*
*Institute."* Yet the list of co-authors further encompasses the majority of
interviewed scientists. I cannot see how this can be objective. I think
interviewees have the right to review their quotes, such that they can verify
that no misunderstandings or misrepresentations have occurred. But I fail to
understand how the interviewees can also be co-author. On the one hand, they
have no access to the other interview transcripts, thus cannot reliably judge if
this was a proper reporting of what was said in the interviews, but more
importantly, as author they have direct impact on which quotes from them are
selected, and how they are presented. That means that this manuscript
basically has become a vehicle to get their own opinions across, which I think
doesn't align with what is expected for a "Research article". On top of that, they
obviously have full access to their own interview, but not to the other
interviews. I cannot see how this can properly result in a good co-authorship,
when the majority of underlying data is inaccessible to the co-author. I cannot
see a scenario where this leads to proper scientific conduct for a peer-reviewed
"Research article". Unfortunately, I don't see how these methodological flaws
can be corrected, and I think the manuscript should be rejected as a peer-
reviewed "Research article". It may find an outlet as an opinion piece.

i)    This quote from Reviewer 1 "*as author they have direct impact on which*
  *quotes from them are selected, and how they are presented. That means*
  *that this manuscript basically has become a vehicle to get their own*
  *opinions across*" somewhat contradicts this one in their epilogue:
  *"Maybe the interviewees expressed themselves somewhat awkwardly*
  *because they also felt like they were in an informal private conversation.*
  *It is also very possible that context or tone went missing in the*
  *transcription and the quote selection for the manuscript"*. If the
  participants "expressed themselves somewhat awkwardly" during the
  interviews, but then could turn the manuscript into a "vehicle to get
  their opinions across", would the participants/co-authors not have, in
  that case, removed any "awkward" quotes?
  In addition, the participants were well aware that the interviews were
  not "informal private conversations". As mentioned L130-131 in the
  manuscript "*participants were emailed with a request for participation*
  *that included a participant information sheet and consent form".* All
  participants had to return the signed consent form prior to being
  interviewed. The consent form states that the interviews were recorded
  and transcribed and that quotes from the interviews may be used in
  future publications.

    *ii)*    As mentioned in the participant information sheet (PIS) and the interview consent form, the methodology used in this study was approved by the University of Edinburgh School of GeoSciences (where CM is based) Research Ethics & Integrity Committee. The Committee consulted the PIS in which it is stated that the participants "*will be given the choice to remain confidential, to be named as a participant or to be a co-author in publications stemming from this study*". The Committee, therefore, concluded that inviting participants to become co-authors did constitute "proper scientific conduct", perhaps because committee members are familiar with qualitative methodologies and knew that it is becoming increasingly customary to invite participants to co-author the research they participated in (e.g. see Given, 2008; Pope, 2020; Farbotko et al, 2021; Doering et al., 2022; Warman et al., 2024)

2. I also struggled with understanding the modeling environment that the authors were considering. I found that the manuscript paints a picture of this environment that simply didn't resonate with me. For example, when I read: [L549-553] *"Yet, models are a product of one or multiple modelers' vision. This was reflected in the interviews during which many participants often mentioned the name of the model creator or lead developer instead of, or as well as, the model's name. The research identity of many modellers is, whether they want it or not, intertwined with their model; inviting authors to reflect about their positionality would allow modelers to regain control over their own narrative and research identity."* My personal experience is completely different. Thinking about the snow model I work with most, and which is widely used and recognized in various cryosphere communities, basically all major model developments in the last 15 years were done by PhD students and PostDocs, most of whom have since moved on. So their "research identities" stretch way beyond "their model". I think when asked, very few of the PhD students would describe the model as "their model". In fact, even though they contributed most significantly to model developments, I doubt they will describe their role as a "modeller". The model I'm mostly familiar with, has almost no dedicated, long-term model developers or code maintainers. The large majority of recent code changes (last 15 years) has been done by people with contracts lasting shorter than a few years. The original "model creators", in the meantime, have taken up different research fields, retired or have taken up other roles in academia. For the model I work with most, no "lead developer" can be identified. Thus, I struggle to agree with this proposed narrative of "model creators" or "lead developers" as well as supposedly the concept of "their model" at face value. It needs to be supported by data and analysis. For example by analyzing model code repositories and investigating how many people contributed how much to the code, and in what role. That would give the necessary underpinning of this narrative. I'm now curious if the model ecosystem I work with is the exception, or the rule. It could also signal a bias in the selection of participants for the interviews.

    • We are glad that the manuscript is making Reviewer 1 reflect on their own experience because it suggests that we have reached one of our goals stated in the Conclusion: the novelty in this paper is that "*it is a reflective exercise which, we hope, will be the start rather than the end point of the conversation*".

However, while Reviewer 1's experience could contribute future conversations
and similar research studies, it does not mean that it cancels the participants'
experience. When participants talked about a snow model, they did often
mention the name of the model's creator e.g. Glen's or the Liston model
(SnowModel), Richard's model (FSM), Marie's model (Ivori), or Dave Lawrence
when mentioning CLM; these are the (qualitative) data we base our analysis on.
The exception was Crocus; although some of its developers were named, no one
was singled out and it was generally referred to as "simply" Crocus.

• *"For example by analyzing model code repositories and investigating how many*
*people contributed how much to the code, and in what role."* This would be
assuming that all model code repositories exist, are well maintained and that
protocols about comments were instigated since the birth of the investigated
models and have been respected since. Based on Menard et al. (2021) who
found that up-to-date and well-maintained model documentation was rare, we
would be reluctant to conduct such an analysis.

3. Further, it is written: [L96-97] *"The aim of this study is to understand why decisions*
*made by modellers all over the world and over the past decades have not led to more*
*(or is it "any"?) progress in Arctic snowpack modelling, …".* Given that my personal
experience is that most development is done by researchers on PhD or other short-
term contracts, I think a lot of issues were mentioned that they have no control over,
like funding or the historical legacy of models. In contrast, very little was reported on
the experiences and choices made by PhD students or other short-term contracted
researchers over the course of their model development efforts. I think it plays a role
here that those researchers seem to be absent from the pool of interviewees.

Thank you for this comment. It is exactly because PhD students and casualised
researchers have no control on funding or the historical legacies of the models that
they were not interviewed. The aim of our study was to interview those who decide or
influence model developments. This is what we meant when we wrote: "*we will*
*investigate the construction of snow models through interviews with the individuals*
*who shape their content*". We recognise that what we meant needs clarifying and will
revise the wording for the next version of the manuscript.

4. I found that the manuscript was lacking context. It feels like it is assumed that the
readers understand the problems with snow models in the Arctic. There is very little
substantiation of these problems (basically restricted to L86-95).

We kept this part of the introduction short because we wanted the participants to
explain, in their own words, what *they* thought were the problems with modelling the
Arctic snowpack. We (CM, SR and IM) did not want to dominate the narrative by
explaining in detail what *we* thought were the problems. We will make this clearer in
the introduction, but also ensure that sufficient information is provided for the reader
to have enough context.

In my opinion, it fails to properly introduce the problems to the reader. Furthermore, I
found context lacking in what the past decades have seen in model development and projects focusing on Arctic snowpacks. In modern-day science, which is highly project-driven, national funding agencies are one of the major sources of funding for model development. There is a lot of emphasis in the manuscript on lack of funding, lack of long-term perspective, focus on other regions than the Arctic, as well as a strong sentiment that these "modellers" supposedly live in their own world.

We find this interpretation misleading and incorrect. We particularly refer Reviewer 1 to 3.2.2 Adaptability e.g. *"Although much literature argues that there is a conflict between academic freedom and solution-based or applied science* (e.g. Henkel, 2005; Winter, 2009; Skea, 2019 etc), *we found instead that adaptability and shifting priorities was integral to the participants"* or *"interdisciplinary collaborations were the key motivation for model development, demonstrating the participants' adaptability"*. For clarity, we will replace "participants" with "modellers" in the above sentence.

I would have expected to read much more about efforts undertaken by the Arctic snow community to support model development. How many proposals did they submit? How many were funded? How much of these funds was allocated for model development? I would have expected to see more hard data on this. Also more concrete information about decision making. I.e., if a proposal contains a modeling component, what model is selected and why? How is decided where to focus energy on model development?

Please see L93-110 above to see how the themes and sections in the manuscript address our aim. Some of these questions were addressed during the interviews, but not all answers are in the manuscript and, when they are, the information is scattered throughout the manuscript and, therefore, may be difficult to decrypt. We will be clearer in the revised version of the manuscript. We also wish to highlight that, while mixing quantitative and qualitative research methods can provide important information, this study is qualitative and therefore the interview transcripts are our "hard data" and quantitative answers were not specifically sought during the interviews

Right now, the manuscript comes across as a lot of complaining and finger-pointing.

The manuscript does not aim to complain or finger-point. It simply describes the research environment in which the participants evolve and which shapes and - given, amongst others, limited data availability and limited funding opportunities - constrains their decision-making. Many quotes reflect a much more collaborative environment than Reviewer 1 believes we describe (e.g. Section 3.2.2, L299-301, L380-385), but we agree that they can seem isolated and lost. We will ensure that explanatory text is clearer.

, but a bit more reporting on one's own activities, including some concrete and objective data on funding, money spent, etc., would be expected given the goal set forth by the authors. Here, for example: [L116-118] *"In these discussions, it became clear that the current snow models fell short in representing all the Arctic snowpack*

*processes needed by project collaborators."* We will provide more detail about what
was needed in CHARTER.

and then expect there to be a free, open-source model that fits one's needs, with
proper documentation and an email address that you can send all your questions to
with efficient response times. That is just an unrealistic expectation. These "unrealistic
expectations" are not described in the manuscript and Reviewer 1's comment is
somewhat misleading. In my opinion "modeling" should be an undertaking done by the
community as a whole, where everyone contributes knowledge, expertise, skills, data,
etc.

5. [L182] *"I'm sick of modelers who think the world is a computer screen"* This quote is a
confirmation for me about the big problem of accessibility to fieldwork, combined with
the "hero"-status attached to fieldwork (Nash et al, 2019). Many research positions
including fieldwork ask for previous fieldwork experience, or, alternatively, "outdoor
experience". Particularly back-country skiers, (alpine) climbers, and hikers have an
edge in securing snow-related fieldwork. And we know that "the outdoors" notoriously
lacks diversity (e.g., Winter et al., 2020, Ho and Chang, 2022). Fieldwork is mostly
accessible for PhD students, or senior scientists with previous fieldwork experience.
Model developers often lack access to participating in fieldwork, and people without
access to fieldwork mostly concentrate on doing modeling work. It's important to note
here that even when possibilities arise, fieldwork is not a safe environment for
everyone (Marín-Spiotta et al., 2020), and that could be prohibitive for participation.
The fact of the matter is that many researchers will never go to the field for a variety of
reasons, which may require rethinking of the status of fieldwork (e.g., Bruun et al.,
2023).

This quote was taken from a conversation during which I (CM) told the participant that
some large-scale modellers had told me anecdotally (i.e. prior to the interviews
conducted for this research) that improving the representation of snow in ESMs was
much less important than improving clouds. The quote was a manifestation of the
participant's frustration with such claims. We will either provide more context or
remove the quote.

We are well aware of the accessibility issues and of the much-needed enormous
progress to make field work more accessible, diverse and inclusive. Had this theme
emerged in our data analysis, we would have addressed it, but it did not. Undertones
of endorsing the hero status did emerge in one conversation and were coded as such,
but in order for codes to be included in the final themes, they had to be identified in
multiple conversations, which, in this instance, was not the case. We will provide this
methodological information in the Methods section in the revised manuscript.

The message delivered in this manuscript is mostly one-directional: [L96-97] *"The aim
of this study is to understand why decisions made by modellers all over the world and
over the past decades have not led to more (or is it "any"?) progress in Arctic snowpack
modelling"*, combined with the statement *"I'm sick of modelers who think the world is a
computer screen"*. There are more than 80 quotes in the manuscript and it is misleading of Reviewer 1 to isolate one quote and to claim that this is our message.
Reviewer 1 expects "proper scientific conduct" (L193 above) to be adopted by the
authors; we expect the same from the reviewers.

6.  I so wish the authors would have written "by the Arctic snow community" instead of
"by modellers". I found this diversity, equity and inclusion aspect overwhelmingly
missing from the manuscript. I will further detail my sentiments here in the "Epilogue"
below. We will reword our aim or/and provide clarification regarding the different uses
of "modeller", aligning with Reviewer 1's Minor comment #3.

Minor comments:

1.  Several statements and wordings are vague.

o  [L96-97] *"The aim of this study is to understand why decisions made by*
 *modellers all over the world and over the past decades have not led to more*
 *(or is it "any"?) progress in Arctic snowpack modelling."* See also my major
 concern #3. I think more effort is needed to document and quantify the
 progress that has been made, such that it can be objectively concluded
 whether or not this constitutes "progress". As it stands, this statement
 carries little weight. In fact, the problems with snow modeling in the Arctic
 are poorly introduced in the manuscript. Only L88-95 discuss this aspect, but
 only very marginally.

 This was already answered. See L292-297 above and L561-567 below of this
 reply.

o  [L294-296] *"When I speak to large scale modellers about rain on snow, the*
 *feedback is usually 'we are aware that something needs to be done, but we*
 *have other priorities and we don't have resources for this'. It's not*
 *straightforward."*
 I think I understand what this is about because of my expertise, but for
 reaching a broader audience, it should be made explicit. Please specify what
 the issues with rain-on-snow are. Is it the precipitation phase separation rain
 vs snow, is it the runoff from a snowpack, is it the formation of ice lenses?
 Also, academia is almost fully project driven, so why not write a proposal or
 provide funding otherwise for a model developer to work on improving the
 "rain-on-snow" problems in a model? I think this also relates to my major
 concern #3, listed above, regarding missing context.

 We will clarify.

o  [L372-373] *"the first thing it would do is alert the modelers to the difficulties*
 *that they have in the Arctic that, in the absence of these evaluations, they*
 *wouldn't even know about…"* Please provide examples. The statement suggests that the interviewee knows about difficulties that the modelers
supposedly don't know about. I deem it inadequate to publish a paper with
statements like that, without sufficient backing up of examples, preferably
using peer-reviewed literature. As mentioned previously, we are interested in
how the opinion of decision-makers - in other words their truth based on
their experience, expertise and perspective - inform their decisions. This
quote is about the need to implement a Tundra-SnowMIP and is consistent
with one of the aims of the previous SnowMIP i.e. ESM-SnowMIP, which was
to "identifying previously unrecognized weaknesses in these models"
(Krinner et al., 2018)

○ [L310-311] *"I mean, the idea that you're going to create an arctic snow model*
*in a PhD is...?!"*

This is an incomplete sentence, and I'm not sure what I need to fill in at the
"...?!". Please add some explanation here.

We will.

○ [L537-538] *"Some users of [our model], they probably don't know what*
*they're doing, and sometimes a paper comes where I say ???"*

Please fill in the "???" here. With my social background, I think I understand
what "???" and "?!" is supposed to indicate, but for non-native English
speakers, I think there is a risk here that they don't get the implicit message.

We will.

2. There were a few quotes that I think are wrong, and I wonder if there should not be an
editorial comment that the statement is deemed inaccurate.

○ For example, looking at the publications involving Crocus over the last 10 years,
I don't think the statement [L237] "But, I mean Crocus, it's an avalanche model,
right?" is accurate.

○ Similarly, [L282-L284] *"[Models] are limiting the number of [snow] layers for*
*computational stability and efficiency so they are not respecting the way in*
*which the snow pack is actually built up i.e. in episodic snowfall events, which*
*will form different layers (...)"*. For models like Crocus and SNOWPACK, it is
trivially easy to avoid a limiting number of snow layers. I think it is important to
make an editorial remark, since otherwise, false information gets propagated.

This was already answered. See L122-159 above.

3. Extensive use of the term "Modeller": I'm not sure the word "modeler" is meaningful.
Even the authors seem to have an ambivalent definition, defining it both as "model
developer" [L127] as well as "with expertise in modeling" [L128]. I think there is a
substantial difference between both. Note that in L132, both SPM and LSM "modelers"
are defined as "model developers". Personally, I think labeling someone as a "modeler"
**often attaches an identity to an individual**, where this is not justified. It also has unclear
meaning. Is it someone who uses the model, or someone who develops for the model,
or is it someone who maintains the model code? Is someone who has used a model once in their research career already a "modeler", or is it someone who uses models in
more than, let's say, 50% of their research? I would rather like to see more exact
wording being used, specifically focusing on the role someone has. Like "model user",
"model developer" or "model maintainer". I think IPCC rightfully avoids the word
modeler (referring to L546). But thinking about roles avoids attaching an identity to a
researcher, while allowing to encapsulate the common situation where researchers can
take up different roles during their career, or even within a single project.

We will provide more exact wording.

4. [L427-428]: *"We argue that efforts to represent Arctic snowpack processes would pave*
*the way in the research areas highlighted below for new interdisciplinary*
*collaborations"*. What follows are three rather specific research directions. Not that I
want to argue about their relevance, it is just missing context why those three are
listed, who has set these priorities? Did this come out of the interviews as well?

They did. We will clarify.

Epilogue

I also would like to stress that the manuscript contained quite some material that to me
came across as somewhat "aggressive". I would like to make the authors aware that it left
me with the impression of a poorly working field, with a lack of communication,
collaboration and a missing cooperative mindset.

We note Reviewer 1's concern.

Below, Reviewer 1 expressed concerns about the manuscript not fostering a healthy,
welcoming, open environment and objects to specific quotes being used. Reviewer 1 also
accuses us of "*heavy cherry-picking*", of making up data ("*these sorts of things apparently*
*have been said in the interviews*"), of misleading the participants ("*Maybe the interviewees*
*expressed themselves somewhat awkwardly because they also felt like they were in an*
*informal private conversation*"), of having no consideration for equ(al)ity, diversity and
inclusion. These are very strong accusations of data falsification, manipulation and selection
i.e. of instances of research misconduct. We hope that Reviewer 1 understands that they
were mistaken, now that we have clarified that (1) these quotes are not presented as
"truths" but as opinions that contribute to informing decisions, (2) these quotes illustrate
the themes that were identified during the thematic analysis, (3) that the themes are about
decision making and therefore serve to answer our research question, and (4) qualitative
data (here the quotes) *are* data and that this is a research paper which followed established
methodologies. As mentioned above, we will revise the manuscript to ensure that this
process is clear to all readers.

We would also like to mention that the review process is not an "open" environment either.
While Copernicus publications are leaders in the peer-review process and have dramatically
improved reviewing by making it open-access, reviewers still can, as is the case for Reviewer

1, remain anonymous; a choice we, of course, respect. Nevertheless, there is a power
imbalance in single-blinded reviews (see e.g. Manchikanti et al., 2015; Parmanne et al.,
2023) and with power comes responsibility. We trust that this responsibility includes not
accusing authors of misconduct until having given them the opportunity to prove otherwise.

Examples:

[L182] *"I'm sick of modelers who think the world is a computer screen"*
In fact, many scientists have no other choice but to focus on modeling, since fieldwork in
polar regions is generally poorly accessible (Nash et al., 2019, Karplus et al., 2022). I know
scientists who would give an arm and a leg to go to the field just once, and probably doing
so would increase the quality of their model development efforts considerably. The phrasing
of this statement suggests that the scientist never considered that they could have made an
effort to bring the *"modelers who think the world is a computer screen"* in closer contact
with the real world, instead of saying that they are "sick" of them. This was already
addressed L366-378 of this reply.

[L184-185] *"The[se] models spend so much time doing things that aren't very important for*
*lots of applications that they're kind of worthless"*
Claiming that work done by fellow scientists is worthless, because it doesn't fit one's own
needs, is detrimental to a healthy, open and welcoming academic atmosphere I think.

We wish to clarify that almost half of the quotes used in this manuscript are from modellers
reflecting on their own practice and community (hence L569 "the novelty here is that it is an
insider's job. It is a reflective exercise"). As mentioned L159 in the manuscript, we decided
not to indicate which quotes came from which group unless necessary to improve
understanding of the context within which they were cited. We understand thanks to
Reviewer 1's comments that we must revise this decision and be clearer about which group
the quotes came from. We hope that it will make it clearer that the manuscript is not a
criticism of modellers, but a reflective process that includes modellers and other members
of the Arctic snow community.
[L537-538] *"Some users of [our model], they probably don't know what they're doing, and*
*sometimes a paper comes where I say ???"*
First of all, I'm not really sure what I have to fill in at the "???", but I assume it is some
negative sentiment. In these cases, reaching out to those users can be of great help to the
users, and would foster exchange of knowledge, and, again, an open and welcoming
academic environment.

We make it clear in Section 3.2.2 Adaptability that modellers do collaborate extensively.
Reviewer 1's comment proposes a solution to an issue that our manuscript identified. As we
wrote in the Conclusion, *we hope that this reflective exercise will be the start rather than the*
*end point of the conversation*. For example, the EDI issues in fieldwork that Reviewer 1

highlighted are only starting to be tackled because recent papers have exposed these issues
and those who want to change the system now have academic papers to back their
initiatives. As highlighted in the Conclusion, we argue that our manuscript serves a similar
purpose. It addresses issues that are well-known but have remained hidden in the literature.
Visibility is key to changing practices and our manuscript contributes to making some of the
issues more visible in order to address them.

[L374-375] *"In my sense, large scale climate modellers aren't sufficiently aware of snow. (…)*
*There are so many people who don't care about that"*
I find this quite the accusation that those people don't care. Please provide evidence that
they don't care, for example from reviews of proposals and/or manuscripts. Did papers in
fact get rejected, because reviewers claim that snow is irrelevant? See L532-538 in the
manuscript for examples provided by other participants of how snow is treated in some
manuscript using large scale models. I'm skeptical that that is the case.

[L96-97] *"The aim of this study is to understand why decisions made by modellers all over*
*the world and over the past decades have not led to more (or is it "any"?) progress in Arctic*
*snowpack modelling, …"*
I understand that the phrasing "(or is it "any"?)" is catchy, but it comes across a bit as
dismissive towards publications from, let's say, the last 10 to 20 years, documenting
improvements in modeling approaches, some of which are cited in the manuscript. I would
strongly encourage more precise wording. We agree. As mentioned, L384-385 of this reply,
the aim will be reworded.

Which objective has not been achieved (yet)? The statement that directly precedes "*The*
*aim of this study* etc" answers this question "*No ESM, so far, simulates these Arctic*
*snowpack processes*". As already stated above, we will ensure that sufficient information is
provided for the reader to have enough context, but we maintain this statement to be
accurate with regards to the representation of the snow profile of Arctic snowpacks, vapour
fluxes and ice crust formation in ESMs. We welcome references from Reviewer 1 that could
inform us otherwise.

All the points below have already been made by Reviewer 1 and addressed by the authors
multiple times. There will, therefore, be no further comments.

Also, this phrasing implies that "modelers" are to blame for the supposedly slow progress. In
fact, the manuscript discusses very few decisions made by "modelers" (interpreted by me
here as model developers). And also in light of the sentences I have listed above, I think this
is unfair. There seems to be a lack of healthy collaboration in the field. I am also aware that
there is also a big issue with accessibility (diversity and inclusion) to fieldwork, that in my
opinion plays a role here.

There are also funding agencies, and hiring decisions that I think are to blame for a lack of
resources for model development. Some of those are addressed in the manuscript, some of those are not. But it would have been better to phrase the aim of the study as: "The aim of
this study is to understand why decisions made by the Arctic snow community all over the
world and over the past decades have not led to more  progress in Arctic snowpack
modelling, ..."

I put this feedback as "Epilogue", because for me, it is not relevant to whether or not the
manuscript could be published as a scientific research article, but I hope the authors
become aware that including statements like these, unfortunately left me with the
impression that the field of Arctic snow is a somewhat unhealthy environment, with some
missing collaborative mindset. In a way, I think it's already a problem that these sorts of
things apparently have been said in the interviews, but maybe this was simply the heat of
the moment. Maybe the interviewees expressed themselves somewhat awkwardly because
they also felt like they were in an informal private conversation. It is also very possible that
context or tone went missing in the transcription and the quote selection for the
manuscript.

One could argue that it may be important to report about such sentiments in the field, since
it can signal problems hindering progress. However, it would require proper context,
including identifying this as a problem, and proposing pathways forward to resolve such
conflicts. I think that the authors should seriously consider the purpose, and effect, of
including statements like these in the manuscript.

In my opinion, it doesn't reflect well on the Arctic snow community, and I refuse to believe
that this is the message the authors wanted to get across.

References:

• Bruun, J. M., & Guasco, A. (2023). Reimagining the 'fields' of fieldwork. Dialogues in
Human Geography, 0(0). https://doi.org/10.1177/20438206231178815

• Yi Chien Jade Ho & David Chang(2022)To whom does this place belong? Whiteness
and diversity in outdoor recreation and education, Annals of Leisure Research, 25:5,
569-582, DOI: 10.1080/11745398.2020.1859389

• Karplus MS, Young TJ, Anandakrishnan S, et al. Strategies to build a positive and
inclusive Antarctic field work environment. Annals of Glaciology. 2022;63(87-
89):125-131. doi:10.1017/aog.2023.32

• Marín-Spiotta, E., Barnes, R. T., Berhe, A. A., Hastings, M. G., Mattheis, A., Schneider,
B., and Williams, B. M.: Hostile climates are barriers to diversifying the geosciences,
Adv. Geosci., 53, 117–127, https://doi.org/10.5194/adgeo-53-117-2020, 2020.

• Nash M, Nielsen HEF, Shaw J, King M, Lea MA, et al. (2019) "Antarctica just has this
hero factor…": Gendered barriers to Australian Antarctic research and remote
fieldwork. PLOS ONE 14(1): e0209983.
https://doi.org/10.1371/journal.pone.0209983

• Winter, P.L.; Crano, W.D.; Basáñez, T.; Lamb, C.S. Equity in Access to Outdoor
Recreation—Informing a Sustainable Future. Sustainability 2020, 12, 124.
https://doi.org/10.3390/su12010124

Brun,, E, Martin,, E. Simon., V, Gendre., C. and Coleou., C.. 1989. An energy and mass model
of snow cover suitable for operational avalanche forecasting. f. Glacial., 35 (121), 333–342.

Doering *et al* 2022 *Environ. Res. Lett.* **17** 065014

Farbotko, C., Watson, P., Kitara, T., & Stratford, E. (2023). Decolonising methodologies:
Emergent learning in island research. *Geographical Research*, 61(1), 96–
104. https://doi.org/10.1111/1745-5871.12519

Given, L. M. (2008). Participants as co-researchers. In The SAGE Encyclopedia of Qualitative
Research Methods (pp. 600-601). SAGE Publications, Inc.,
https://doi.org/10.4135/9781412963909

Manchikanti, L., Kaye, A. M., Boswell, M. V., & Hirsch, J. A. (2015). Medical journal peer
review: process and bias. Pain physician, 18(1), E1.

Parmanne, P., Laajava, J., Järvinen, N. *et al.* Peer reviewers' willingness to review, their
recommendations and quality of reviews after the Finnish Medical Journal switched from
single-blind to double-blind peer review. *Res Integr Peer Rev* **8**, 14 (2023).
https://doi.org/10.1186/s41073-023-00140-6

Pope, E. M. (2020). From Participants to Co-Researchers: Methodological Alterations to a
Qualitative Case Study. The Qualitative Report, 25(10), 3749-3761.
https://doi.org/10.46743/2160-3715/2020.4394

Sargeant J. Qualitative Research Part II: Participants, Analysis, and Quality Assurance. J Grad
Med Educ. 2012 Mar;4(1):1-3. doi: 10.4300/JGME-D-11-00307.1. PMID: 23451297; PMCID:
PMC3312514.

Warman, R., Watson, P., (Amy) Lin, C.C., Allen, P., Beazley, H., Junaidi, A. et al. (2024) A
labour of love: Cross-cultural research collaboration between Australia and Indonesia. *Geo:*
*Geography and Environment*, 11, e00132. Available from: https://doi.org/10.1002/geo2.132

All other references can be found in the manuscript.

---

## Author Comment (AC2)

We thank Reviewer 2 for supporting this manuscript and for providing us with detailed comments. We also thank them for deepening our knowledge on philosophical literature on decision-making in modeling. Having now read the suggested papers, we agree that they are important to our paper and that they will help frame our discussions further.

This paper covers an important topic—how decisions are made in the context of modeling of Arctic snowpack. By providing insight into the influences on decisions throughout the modeling process the paper contributes to a minimally understood feature of modeling practice, as in most cases, the decision-making is not explicitly documented, nor are the reasons that justify particular decisions. However, there are some issues with the manuscript that need to be addressed:

I urge the authors to consider some of the philosophical literature on decision-making in modeling, which mainly concerns climate models but applies to the discussions and perhaps the interpretations of some of these qualitative findings.

1. Several philosophers working on issues in climate science have detailed how values (i.e., interests) influence decision-making through the course of model development, but none of that literature is referenced here despite the high relevance to the topic of discussion. I recommend looking at Parker and Winsberg (2018), Parker (2014), and Morrison (2021), specifically chapter 3 of the latter. The research by these scholars discusses how interests (subjective preferences) and features of the modeling context (pragmatics) influence decision-making in the course of climate modeling, including those choices of determining modeling purposes and priorities, what and how to represent features of the target system, the suitability of observations and metrics for model assessment and validation, etc. The authors might also consider looking at the Shackley article "Epistemic Lifestyles in Climate Change Modeling" (2001). I suggest adding elements from these papers to the paragraph starting in line 64 or including an additional paragraph to capture the discussions in philosophy on these topics. You might also find that the insights from certain papers are relevant to specific sections as well (for example, Parker 2014 for section on data available and resources.

We will add these references throughout our paper. We will add a subsection in Section 4 that will frame our findings within topics discussed in these and other papers suggested below by Reviewer 2.

2. And, concerning tradeoffs, see work by Levins, mainly "The Strategy of Model-building in population biology" (1966). I note that the subject of modeling is different, but Levins' thesis applies to the modeling of complex systems generally and is thus related to the discussion in 3.1.1.

We will consider Levins' strategies in the new subsection in Section 4.

Concerning the disagreement about models being "good enough" for current research problems—an article deals with similar disagreement about the value of different modeling systems in relation to different sets of research questions by Lloyd, Bukovsky, and Mearns (2020). The authors here argue that the reason for disagreements about the value of regional versus global models is because they have different research questions and the representational features of the models are different. So they don't take the representational features of one type of model to be valuable for their questions, and vice versa. Wonder if something similar here is going on, thus this frame might be useful…and might even be useful for analyzing the lack of unanimity in the responses to the questions that were asked. They have different interests, are asking different questions, and have different local epistemologies (Longino 2002 and Morrison 2021). (Where the authors talk about identity, this seems akin to
local epistemologies.)

We will add references and discussions by philosophers of science in the additional subsection
to help reframe our discussion.

Regarding identity and local epistemologies (LE): We agree that considerations about LE are
relevant to the paper and this will be considered in the next version of the manuscript.
However, we draw our analysis on research identity from numerous studies on academic and
research identities in the field of education studies (e.g. Valimää,1998; Clegg, 2008;
Fitzmauritz, 2013; Borluag et al., 2023). For example, what philosophers of science call LE are
akin to disciplinary identity in education studies (e.g. Dressen-Hammouda, 2008). Therefore,
rather than being aking to identity, we believe that LE are *part of* identity construction, i.e. they
are the "*processes of identification with diverse groups and communities*" in the McCune
(2019) definition quoted in the manuscript, as are values. In addition, considering our findings
in terms of LE would imply that all participants within the same group would agree (as per the
examples in the cited papers "regional" vs "global" climate modellers), which is not the case.
There was a lot of within group disagreement, which we will made clearer in the next version
of the manuscript by attributing quotes to specific groups.

Appreciate the content-context distinction, however, I wonder if you can separate them, and would
appreciate more consideration of the way research context, understood more generally than
"identity" in the paper, shapes perception of modeling practice, etc.

Please see above regarding "identity".

Regarding the content-context distinction, we believe it is necessary, because, as written in the
conclusion of the paper, "*while the written history narrated by our publications does record the*
*arguments presented here in the content section, it does not record what is presented in the context*
*section*". Our paper was submitted to The Cryosphere (TC), a journal which, as far as we are aware,
has never published a research paper based on qualitative methodologies. We chose to submit the
paper to TC because the TC readership is the audience we want to engage with our paper because,
as written L569-570 of the paper "*it is an insider's job. It is a reflective exercise which, we hope, will*
*be the start rather than the end point of the conversation*". As shown by Reviewer 1's comments, we
must expect that some of the readership will be unfamiliar with these methodologies, therefore we
must ensure that they recognise some of the findings (Content) or they may disengage with the
broader discussion (Context and Moving forward).

I am also not sure whether the analysis from Staddon (2017) on the distinction between professional
and personal is fitting here

We agree. The Staddon excerpt was echoed in Section 4.3 "Values and positionality" where it was
referenced again, although not explicitly "*Values are another construct to a researcher's identity, but*
*the prevailing notion linking value-free science with objectivity and impartiality (Pulkkinen et al.,*
*2021) presents obstacles to achieving greater transparency in bridging the gap between our personal*
*identities and our professional decisions.* ". We will either reference it more explicitly the second
time or remove it.

Again, I think these responses are a function of differences in the context in which these individuals
conduct research and the local epistemologies they are part of. For example, with "I'm sick of modelers who think the world is a computer screen" this is a rejection of the attitude of being
focused on the modeling world as opposed to the empirical world, which can be reduced to
differences in one's scientific ontology and epistemic values. And "these models spend so much
time…" this can be interpreted as someone who is more of a pluralist about models and their
application, as opposed to part of the paradigm by which models are seen as fit-for-purpose for a
limited number of intentionally chosen applications….in other words, it's not necessarily the
"identities" of the researchers that come out in these quotes, but rather, the diversity of local
epistemologies that can be found in Arctic modeling, and the disagreement that arises from this
diversity. I appreciate the information in the intro to section 3 but think you could do more to shed
light on the significance of sharing these sorts of quotes from your interviews. A different frame for
your discussion might add depth and significance.

We agree with Reviewer 2. We will return to the quotes used in the intro to section 3 in Section 3.2
or 4 and will frame them within a broader discussion about the constructs of researcher identity
(which include values and local epistemologies; see above for details).

In the same vein as the above comments, I think philosophical discussions can help to frame your
results. For example, the somewhat reductive interpretation of the quote at the beginning of 3.2.1.:
prioritization is a feature of scientific practices, including modeling, being driven by human interests,
and certain elements of the complex systems we investigate being more or less important relative to
those interests. While resources are limited, human beings are also inherently value-driven, and if
they don't perceive something as related to their interests, they will deprioritize it, and yes, the
practical constraints make this more apparent, but aren't the sole cause of prioritization in science.
There are an infinite number of questions we could ask, and we will see value in some and ignore
others. I think this is what the quote is getting at you have chosen here, with the "we have other
priorities" AND "we don't have resources", i.e., there are two reasons for not tackling the problem,
one is, it is inconsistent with what they care about in modeling, and second, there aren't resources,
and these compound one another. Longino's discussions of modeling complex systems in her 2002
book would be helpful here. This is an example of one place in the manuscript where the
interpretation of qualitative evidence can be aided by appealing to philosophical discussions from
the philosophy of science in practice (i.e., Longino and others have done empirical studies to draw
their conclusions, it's not "armchair" analysis).

We agree with Reviewer 2. These considerations will be addressed in the new additional subsection.

The comments on short-termism are incredibly important, appreciate their explicit inclusion, and
wonder if more can be said about the implications of this current paradigm in funding procedures…

We will provide more context around the comments on short-termism.

I am a bit confused about the discussion of the anchoring bias…it appears a bit vague in what the
bias is in itself, and I am not sure that the explanation in the first paragraph makes it clear what it is.
I think it is the judged adequacy of the models, based on historical model features and development,
in relation to some purpose, which can shift when one's interests or research questions change
(which the authors hint at in lines 375–379). I think this is what is being said also in the case that
community efforts can lead to shifts in these anchors…community comparison projects foster
interdisciplinary discourse on model capabilities and limitations, which can presumably highlight
inadequacies in relation to priority research questions. This section could be clearer, especially with
respect to what it is about the existing models that function as a reference point for judging the
value of different future development efforts. The section should also conclude with a clear summary of the argument the authors seek to make given the statement in the first paragraph:
"anchoring contributed largely to the absence of Arctic snow processes in existing models".

The interpretation of Reviewer 2 is correct and we will clarify this in subsection 3.2.3.

In conclusion, this is a valuable study and provides significant empirical insight into understudied and
implicit components of modeling of climate features generally. However, I think work needs to be
done with the framing of the findings from the study and their discussion. I strongly suggest bringing
in philosophical work on modeling to help add depth and detail to the discussion.

References:

Levins, R. (1966). The strategy of model building in population biology. *American scientist*, *54*(4),
421-431. (see also, for updated discussions: Weisberg, M. (2006). Forty years of 'the strategy': Levins
on model building and idealization. *Biology and Philosophy*, *21*, 623-645. and Matthewson, J. (2011).
Trade-offs in model-building: A more target-oriented approach. *Studies in History and Philosophy of*
*Science Part A*, *42*(2), 324-333.)

Lloyd, E. A., Bukovsky, M., & Mearns, L. O. (2021). An analysis of the disagreement about added
value by regional climate models. *Synthese*, *198*(12), 11645-11672.

Longino, H. E. (2002). *The fate of knowledge*. Princeton University Press. (See chapter 8 for local
epistemologies and differences between different investigative communities, which is relevant to
your discussion.)

Morrison, M. A. (2021). *The models are alright: A socio-epistemic theory of the landscape of climate*
*model development*. Indiana University.

Parker, W. (2014). Values and uncertainties in climate prediction, revisited. *Studies in History and*
*Philosophy of Science Part A*, *46*, 24-30.

Parker, W. S., & Winsberg, E. (2018). Values and evidence: how models make a difference. *European*
*Journal for Philosophy of Science*, *8*, 125-142.

Shackley, S. (2001). Epistemic lifestyles in climate change modeling.
**Citation**: https://doi.org/10.5194/egusphere-2023-2926-RC2

Borlaug, S.B., Tellmann, S.M. & Vabø, A. Nested identities and identification in higher education
institutions—the role of organizational and academic identities. *High Educ* 85, 359–377,
https://doi.org/10.1007/s10734-022-00837-5, 2023.

Clegg, S., Academic identities under threat?, *British Educational Research Journal*, 34, 329-345,
https://doi.org/10.1080/01411920701532269, 2008.

Dressen-Hammouda, D., From novice to disciplinary expert: Disciplinary identity and genre mastery,
*English for Specific Purposes*, 27, 233-252, https://doi.org/10.1016/j.esp.2007.07.006, 2008.

Fitzmaurice, M., Constructing professional identity as a new academic: a moral endeavour, *Studies in*
*Higher Education*, 38, 613-622, DOI: 10.1080/03075079.2011.594501, 2013.

Välimaa, J., Culture and Identity in Higher Education Research. *Higher Education*, 36, 119–138, 1998.

---

## Referee Report (RR1)

**Main comments:**

The comments of the reviewers have, on my view, mostly been addressed well. There are, however, two related ways in which I think the paper still needs work. These relate somewhat to reviewer 1's worries about the methodology of the paper.

According to the paper's introduction, the paper aims to "understand why decisions made by the snow modelling community over the past decades have led to little or no progress in the representation of Arctic snowpack processes, i.e. in the part of the planet that warms faster than anywhere else." No mention is made of further goals of the paper in either the introduction or the methodology section. But section 4's aim is to draw conclusions from the study about what snow modelling should continue to do and what it should start to do. While the issue with the paper here is partly structural, in that the reader needs advance notice about section 4, the issue is also methodological. What exactly is the basis for the recommendations in section 4? How do the recommendations relate to the material in earlier sections? What are the limitations of the recommendations? What is novel about the recommendations? These questions are hard for the reader to answer, partly because of the absence of information in the methodology section, partly because the arguments offered in section 4 are quick, and partly because the authors suggest that they do not assume the correctness of the opinions of those interviewed while some arguments seem to rest on these opinions (see the detailed comments on lines 583-591, 609-610, and 613-622).

The structural issue should, I propose, be addressed in the introduction. The methodological issue should be addressed in the methodology section. The conclusions should also include some clarification relating to the scope of the claims and their basis. I was particularly unclear about the argument for a plurality of modelling approaches.

**Detailed comments:**

Line 98: 'i.e.' seems inappropriate here. More importantly, the claim that perhaps no progress has occurred in modelling snowpack seems too strong because the previous paragraph does describe some progress and only states that there has been no progress in ESMs. Perhaps an additional sentence is needed in the previous paragraph to make it more explicit that most models still represent no snow processes (as recognised on the next page). This would help understand the claim that perhaps no progress has been made.

Line 154: Could you be more explicit about how the expertise of the extra four participants helped to address the existence of overlap in expertise?

Lines 208 and 238: The references are not entered properly here.

Line 286: "*Everything always start at…*" Missing 's'.

Line 340: The reference is not entered properly here.

Line 440-441: it is stated that the paper argues that anchoring is an important reason for the absence of Arctic snow processes in existing models. This seems out of place here, given that all you are doing in this section is reporting on the interviews. I would suggest deleting this sentence.

Lines 467-469: this sentence is hard to parse. I suggest reformulating.

Line 569: the statement about being sick of certain modellers is mentioned for the second time. This is a bit jarring because it is presented as if it is mentioned for the first time.

Line 583-591: the argument here for a plurality of modelling strategies was hard to follow and seems too quick. Is the idea just that the system is complex so multiple perspectives are needed? I would have expected more detail than this.

Lines 609-610: the paper here seems to be appealing to the opinions gathered in the interviews (reported in section 4.1) to make proposals for future modelling. Is this correct? Why is this appeal allowed here? My impression was that the paper was not supposed simply to endorse these.

Lines 613-622: the argument here about running future MIPs seems partly to appeal to the opinions of those interviewed. Is this permitted here?

Line 616: reference not entered properly here.

Line 676: "written it" should be "written".

---

## Author Response (AR2)

We thank the reviewer for their considerate comments.

**Main comments:**

The comments of the reviewers have, on my view, mostly been addressed well. There are, however,
two related ways in which I think the paper still needs work. These relate somewhat to reviewer 1's
worries about the methodology of the paper. According to the paper's introduction, the paper aims
to "understand why decisions made by the snow modelling community over the past decades have
led to little or no progress in the representation of Arctic snowpack processes, i.e. in the part of the
planet that warms faster than anywhere else." No mention is made of further goals of the paper in
either the introduction or the methodology section. But section 4's aim is to draw conclusions from
the study about what snow modelling should continue to do and what it should start to do. While
the issue with the paper here is partly structural, in that the reader needs advance notice about
section 4, the issue is also methodological.

In order to avoid confusion, we have clarified that Section 4 is the discussion section and, as such,
renamed it "Discussion and moving forward". We also changed the introductory text to Section 4.
Some of the purposes that a discussion serves are to discuss and synthesize the findings, to situate
the findings within a broader research context and to recommend future research: this is exactly
what we did in Section 4 (also the answer below). As such, we do not agree that the issue is
methodological – on the contrary, we followed the well-established conventions of the structure of a
research paper.

As requested by the reviewer, we now mention in the Introduction that "*The underlying premise of*
*th[e] aim is rooted in the belief that comprehending the cause of a problem – if indeed the absence of*
*an Arctic snowpack is one – provides a foundation for addressing it and recommending ways to move*
*forward, which we will do in the Discussion section*."

What exactly is the basis for the recommendations in section 4? How do the recommendations
relate to the material in earlier sections? What are the limitations of the recommendations?* What
is novel about the recommendations?* See below (*) for responses to these questions. These
questions are hard for the reader to answer, partly because of the absence of information in the
methodology section, partly because the arguments offered in section 4 are quick,

We hope that the changes mentioned above will have partly addressed these questions. All the
recommendations and "*arguments offered in section 4*" are, as they should in any discussion, drawn
from the findings. This is made clear throughout as every subsection in Section 4 clearly references
which findings they draw from and reflect on. Please see the full text, but a selection of quotes from
each subsection can be found below:

• Section 4.1: "*As mentioned in Section 3.2.3, values have contributed to deciding priorities…*";
"*As mentioned in Section **Error! Reference source not found.**, SMC, FS, and RSS were*
*interviewed to provide a broad picture of the range of Arctic snow applications…*"
• Section 4.2 "*The different opinions expressed throughout this paper suggest that the*
*participants support different strategies*"; "*As noted in Sections 3.2.2 and 4.1, collaborations*
*are drivers for model developments…*"
• Section 4.3: "*if a tundra SnowMIP is to advance snow modelling, the obstacles that limit the*
*implementation of Artic tundra snow processes (see subsections **Error! Reference source not***
***found.**.x) should be directly addressed*". MIPs and SnowMIP are also mentioned multiple
times during the manuscript.

- Section 4.4: *"Participants in this study have provided various reasons for not having*
*prioritised the development of an Arctic snowpack model"*; *"This was reflected in the*
*interviews during which more participants referred to Richard's model, Glen's model or*
*Marie's model rather than to FSM, SnowModel and IVORI respectively"*.  This section also
reflects on the methodology used and discussed in Section 2.

* What are the limitations of the recommendations? What is novel about the recommendations?

The entire Conclusion explicitly addresses these questions, but below is a summary:  *"As per more*
*conventional review papers, the novelty in this paper is not in its content, but in the medium it*
*chooses to present that content.(…) In fact, the medium is not novel either. Science and technology*
*studies examine the context within which science is constructed and philosophers of science have*
*long debated the decision-making process of scientists. (…) Therefore, the novelty here is that it is an*
*insider's job. It is a reflective exercise which, we hope, will be the start rather than the end point of*
*the conversation"*.

and partly because the authors suggest that they do not assume the correctness of the opinions of
those interviewed while some arguments seem to rest on these opinions (see the detailed
comments on lines 583-591, 609-610, and 613-622).

What we wrote was *"none of the quotes are endorsed by all authors and, by extension, it is expected*
*that readers will also inevitably disagree with some quotes"*. This remains true throughout the paper
and if *"none of the quotes are endorsed by all authors"*, inevitably some quotes will be endorsed by
some authors. Given the breadth of opinions provided by the participants, it is inevitable that *"some*
*arguments* [do] *rest on these opinions"*. By continuing to quote the participants in the Discussion
section, we aim to give credit to their contributions and to show how their perspectives have,
inevitably, informed this part of the manuscript.

The structural issue should, I propose, be addressed in the introduction. The methodological issue
should be addressed in the methodology section. The conclusions should also include some
clarification relating to the scope of the claims and their basis. I was particularly unclear about the
argument for a plurality of modelling approaches. These points are individually addressed in other
parts of this response.

**Detailed comments:**

Line 98: 'i.e.' seems inappropriate here. More importantly, the claim that perhaps no progress has
occurred in modelling snowpack seems too strong because the previous paragraph does describe
some progress and only states that there has been no progress in ESMs. Perhaps an additional
sentence is needed in the previous paragraph to make it more explicit that most models still
represent no snow processes (as recognised on the next page). This would help understand the
claim that perhaps no progress has been made.

As suggested by the reviewer, we added text to an existing sentence, to make it more explicit that
most models still do not represent Arctic snowpack processes:

Original sentence: "No ESM, so far, does".

New sentence: "However ,no ESM, i.e. none of the state of the art models that are used by
researchers and policymakers globally to understand the complex interactions in the Earth's climate
system, so far, simulates Arctic-specific snowpack processes".

Line 154: Could you be more explicit about how the expertise of the extra four participants helped to
address the existence of overlap in expertise? Done.

Lines 208 and 238: The references are not entered properly here.

Line 286: "Everything always start at…" Missing 's'. Done

Line 340: The reference is not entered properly here. Thank you.

Line 440-441: it is stated that the paper argues that anchoring is an important reason for the
absence of Arctic snow processes in existing models. This seems out of place here, given that all you
are doing in this section is reporting on the interviews. I would suggest deleting this sentence. We
removed the sentence and made the start of the section clearer.

Lines 467-469: this sentence is hard to parse. I suggest reformulating. The sentence was
reformulated.

Line 569: the statement about being sick of certain modellers is mentioned for the second time. This
is a bit jarring because it is presented as if it is mentioned for the first time. The version of the quote
in this section is longer than the version used earlier, which is why it was not referenced. We added
"as mentioned in Section 3".

Line 583-591: the argument here for a plurality of modelling strategies was hard to follow and seems
too quick. Is the idea just that the system is complex so multiple perspectives are needed? I would
have expected more detail than this.
There is more detail. L583-591 are the conclusion to the entire section, which explains the history to
the plurality of modelling strategies (indeed, the "idea [is] that the system is complex so multiple
perspectives are needed"), how epistemologists have related it to climate science and finally, how
the concept relates to our findings. We have rewritten this paragraph to make this clearer.
Lines 609-610: the paper here seems to be appealing to the opinions gathered in the interviews
(reported in section 4.1) to make proposals for future modelling. Is this correct? Why is this appeal
allowed here? My impression was that the paper was not supposed simply to endorse these. Please
see L60-66 above.

Lines 613-622: the argument here about running future MIPs seems partly to appeal to the opinions
of those interviewed. Is this permitted here? Please see L60-66 above.

Line 616: reference not entered properly here. Thank you.

Line 676: "written it" should be "written". Done